# A postsynaptic PI3K-cII dependent signaling controller for presynaptic homeostatic plasticity

Anna G Hauswirth[1,3], Kevin J Ford[1,3], Tingting Wang[1,3], Richard D Fetter[1,3], Amy Tong[1,3], Graeme W Davis[1,3]*

[1]Department of Biochemistry and Biophysics, University of California, San Francisco, San Francisco, United States; [3]Kavli Institute for Fundamental Neuroscience, University of California, San Francisco, San Francisco, United States

**Abstract** Presynaptic homeostatic plasticity stabilizes information transfer at synaptic connections in organisms ranging from insect to human. By analogy with principles of engineering and control theory, the molecular implementation of PHP is thought to require postsynaptic signaling modules that encode homeostatic sensors, a set point, and a controller that regulates transsynaptic negative feedback. The molecular basis for these postsynaptic, homeostatic signaling elements remains unknown. Here, an electrophysiology-based screen of the *Drosophila* kinome and phosphatome defines a postsynaptic signaling platform that includes a required function for PI3K-cII, PI3K-cIII and the small GTPase Rab11 during the rapid and sustained expression of PHP. We present evidence that PI3K-cII localizes to Golgi-derived, clathrin-positive vesicles and is necessary to generate an endosomal pool of PI(3)P that recruits Rab11 to recycling endosomal membranes. A morphologically distinct subdivision of this platform concentrates postsynaptically where we propose it functions as a homeostatic controller for retrograde, trans-synaptic signaling.
DOI: https://doi.org/10.7554/eLife.31535.001

## Introduction

Homeostatic signaling systems stabilize the functional properties of individual neurons and neural circuits through life (*Marder and Prinz, 2002*; *Turrigiano et al., 1998*; *Burrone et al., 2002*; *Davis, 2006*, *2013*). Despite widespread documentation of neuronal homeostatic signaling, many fundamental questions remain unanswered. For example, given the potent action of homeostatic signaling systems, how can neural circuitry be modified during neural development, learning, and memory? Although seemingly contradictory, the homeostatic signaling systems that stabilize neural function throughout life may actually enable learning-related plasticity by creating a stable, predictable background upon which learning-related plasticity is layered (*Davis, 2013*; *Keck et al., 2017*). Therefore, defining the underlying molecular mechanisms of homeostatic plasticity may not only inform us about the mechanisms of neurological disease, these advances may inform us regarding how complex neural circuitry is able to accomplish an incredible diversity of behaviorally relevant tasks and, yet, retain the capacity for life-long, learning-related plasticity.

Neuronal homeostatic plasticity encompasses a range of compensatory signaling that can be subcategorized based upon the cellular processes that are controlled including ion channel gene expression, neuronal firing rate, postsynaptic neurotransmitter receptor abundance and presynaptic vesicle release (*Anggono et al., 2011*; *Burrone et al., 2002*; *Davis, 2006*, *2013*; *Haedo and Golowasch, 2006*; *Maffei and Fontanini, 2009*; *Marder and Prinz, 2002*; *Parrish et al., 2014*; *Stellwagen and Malenka, 2006*; *Turrigiano et al., 1998*; *Watt and Desai, 2010*; *Zhang et al., 2003*). Presynaptic homeostatic potentiation (PHP) is an evolutionarily conserved form of neuronal

*For correspondence:
graeme.davis@ucsf.edu

homeostatic control that is expressed at the insect, rodent and human neuromuscular junctions (NMJ) (*Cull-Candy et al., 1980*; *Davis and Müller, 2015*; *Frank et al., 2006*; *Plomp et al., 1995*) and has been documented at mammalian central synapses (*Burrone et al., 2002*; *Kim and Ryan, 2010*; *Liu and Tsien, 1995*). PHP is initiated by the pharmacological inhibition of postsynaptic neurotransmitter receptors. The homeostatic enhancement of presynaptic vesicle release can be detected in a time frame of seconds to minutes, at both the insect and mouse NMJ (*Frank et al., 2006*; *Wang et al., 2016*). This implies the existence of postsynaptic signaling systems that can rapidly detect the disruption of neurotransmitter receptor function and convert this into retrograde, transsynaptic signals that accurately adjust presynaptic neurotransmitter release (*Figure 1A*; see also for review *Davis, 2006*, *2013*; *Müller et al., 2015*). Notably, the rapid induction of PHP is transcription and translation independent (*Goold and Davis, 2007*), calcium-independent (*Frank et al., 2009*), and does not include a change in nerve terminal growth or active zone number (*Frank et al., 2006*, *2009*; *Harris et al., 2015*; *Younger et al., 2013*; *Wang et al., 2016*).

There has been considerable progress identifying presynaptic effector molecules responsible for the expression of PHP (*Dickman and Davis, 2009*; *Frank et al., 2009*; *Harris et al., 2015*; *Müller and Davis, 2012*; *Müller et al., 2015*; *Wang et al., 2016*). There has also been progress identifying postsynaptic signaling molecules that control synaptic growth at the *Drosophila* NMJ (*Ballard et al., 2014*; *Chen et al., 2012*; *DiAntonio et al., 2001*; *Harris et al., 2016*; *McCabe et al., 2004*) as well as the long-term, translation-dependent maintenance of PHP (*Goold and Davis, 2007*; *Kauwe et al., 2016*; *McCabe et al., 2004*; *Penney et al., 2012*, *2016*). However, to date, nothing is known about the *postsynaptic* signaling systems that initiate and control the rapid induction and expression of PHP.

Here, we report the completion of an unbiased, forward genetic screen of the *Drosophila* kinome and phosphatome, and the identification of a postsynaptic signaling system for the rapid expression of PHP that is based on the activity of postsynaptic Phosphoinoside-3-Kinase (PI3K) signaling. There are three classes of PI3-Kinases, all of which phosphorylate the 3 position of phosphatidylinositol (PtdsIns). Class I PI3K catalyzes the conversion of PI(4,5)P$_2$ to PI(3,4,5)P$_3$ (PIP3) at the plasma membrane, enabling Akt-dependent control of cell growth and proliferation (*Carracedo and Pandolfi, 2008*; *Vanhaesebroeck et al., 2012*), and participating in the mechanisms of long-term potentiation (*Knafo and Esteban, 2012*). Class II and III PI3Ks (PI3K-cII and PI3K-cIII, respectively) both catalyze the conversion of PI to PI(3)P, which is a major constituent of endosomal membranes. PI(3)P itself may be a signaling molecule with switch like properties, functioning in the endosomal system as a signaling integrator (*Zoncu et al., 2009*). The majority of PI(3)P is synthesized by PI3K-cIII, which is involved in diverse cellular processes (*Backer, 2008*; *Dall'Armi et al., 2013*). By contrast, the cellular functions of PI3K-cII remain less well defined. PI3K-cII has been linked to the release of catecholamines (*Meunier et al., 2005*), immune mediators (*Nigorikawa et al., 2014*), insulin (*Dominguez et al., 2011*), surface expression and recycling of integrins (*Ribeiro et al., 2011*), and GLUT4 translocation to the plasma membrane, a mediator of metabolic homeostasis in muscle cells (*Falasca et al., 2007*). Here, we demonstrate that Class II and Class III PI3K-dependent signaling are necessary for the rapid expression of PHP, controlling signaling from Rab11-dependent, recycling endosomes. By doing so, we define a postsynaptic signaling platform for the rapid expression of PHP and define a novel action of PI3K-cII during neuronal homeostatic plasticity. To our knowledge, this is the first established postsynaptic function for PI3K-cII at a synapse in any organism.

Recently, it has become clear that the endosomal system has a profound influence on intracellular signaling and neural development. There is evidence that early and recycling endosomes can serve as sites of signaling intersection and may serve as signaling integrators and processors (*Irannejad et al., 2015*; *Villaseñor et al., 2016*). Furthermore, protein sorting within recycling endosomes, and novel routes of protein delivery to the plasma membrane, may specify the concentration of key signaling molecules at the cell surface (*Choy et al., 2014*; *Hanus and Ehlers, 2016*; *Issman-Zecharya and Schuldiner, 2014*; *Solis et al., 2017*). The essential role of endosomal protein trafficking is underscored by links to synapse development (*Lloyd et al., 2002*; *Seto et al., 2002*) and neurodegeneration (*Pennetta et al., 2002*; *Sanhueza et al., 2014*). Yet, connections to homeostatic plasticity remain to be established. Based upon the data presented here and building upon prior work on endosomal signaling in other systems, we speculate that postsynatpic PI3K-cII and Rab11-dependent recycling endosomes serve as as a postsynaptic 'homeostatic controller' that is essential for the specificity of retrograde, transsynaptic signaling.

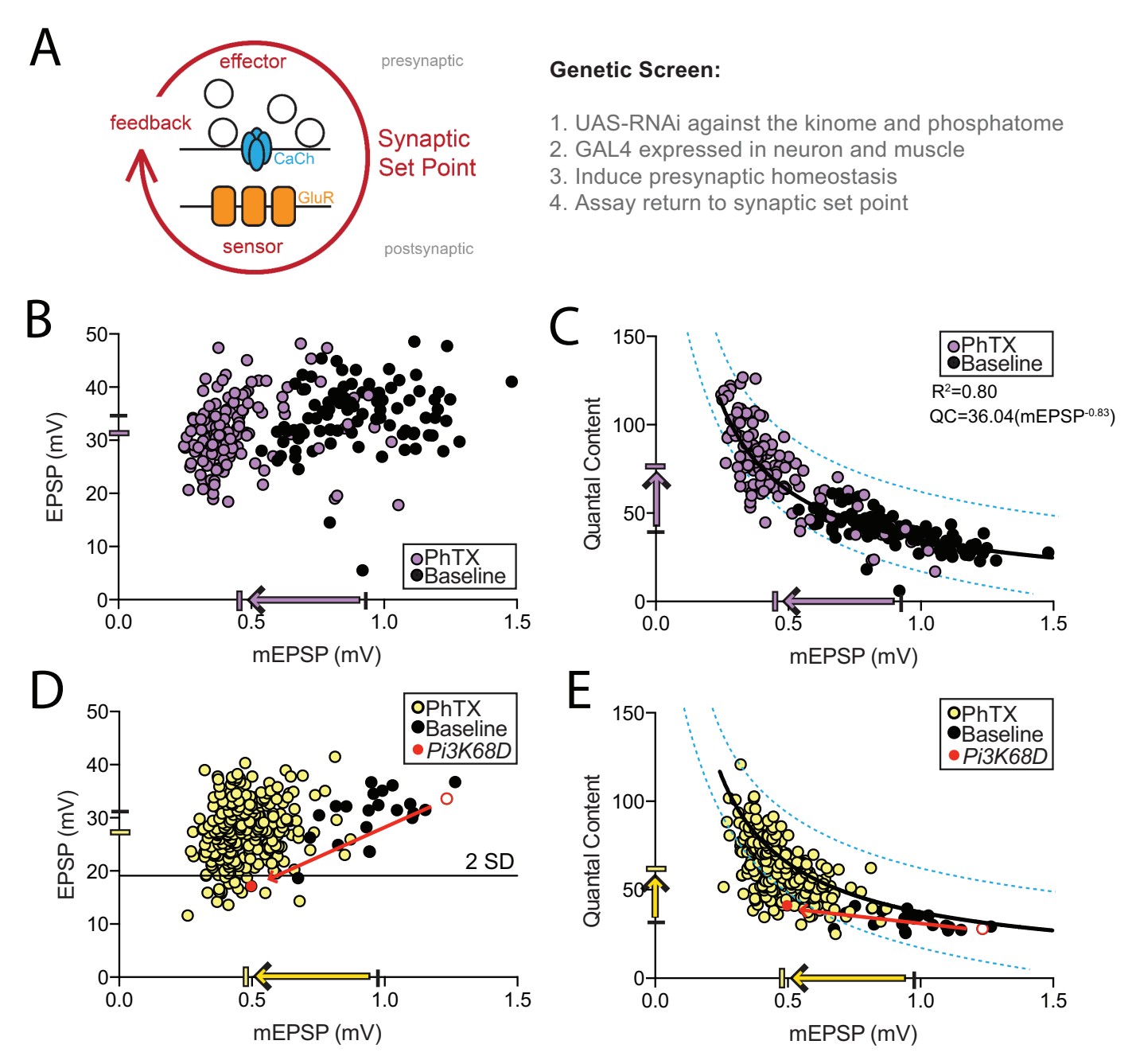

**Figure 1.** Screen of *Drosophila* kinome and phosphatome in presynaptic homeostatic plasticity. (A) Schematic highlighting the trans-synaptic nature of presynaptic homeostatic plasticity. The 'synaptic set point' is operationally defined as the combined action of all pre- and postsynaptic parameters that reliably specify the transfer of information at the synapse. The synaptic set point is stably maintained by a trans-synaptic, homeostatic signaling circuit (red arrow) that includes postsynaptic sensors, retrograde feedback signaling, and presynaptic effectors that drive changes in presynaptic vesicle release. At right, the forward genetic screen of the *Drosophila* kinome and phosphatome is outlined. (B–C) Each point represents average data from a single NMJ. Purple points are in the presence of PhTX. Black points are in the absence of PhTX (baseline). The black hash marks on X and Y axes designate average mEPSP and EPSP amplitudes, respectively, without PhTX. The purple hash marks are averages in the presence of PhTX. The black line in (C) is a power curve fit, equation indicated on graph. The dotted blue lines encompass 95% of all wild-type data points. Recordings made at 0.35 mM $[Ca^{2+}]_e$. (D–E) Screen data of kinase and phosphatase *UAS-RNAi* driven by muscle and neuron *GAL4* plotted as in (B–C) except that each point represent genotypic averages, yellow points (+PhTX) and black circles (-PhTX, baseline). The black line in (D) denotes two standard deviations below the population mean EPSP amplitude of control genotypes from (B). In (D) and (E) the red dot represents *GAL4, UAS-RNAi* for *Pi3K68D*.The black line in (E) is the same curve fit to the control data set in C, layered onto the experimental screen data for comparison. Blue dotted lines as in (C). Recordings made at 0.35 mM $[Ca^{2+}]_e$.

*Figure 1 continued on next page*

*Figure 1 continued*

DOI: https://doi.org/10.7554/eLife.31535.002

The following figure supplement is available for figure 1:

**Figure supplement 1.** Quantification of additional screening parameters.

DOI: https://doi.org/10.7554/eLife.31535.003

## Results

There are at least 251 kinases and 86 phosphatases, either functionally annotated or predicted, in the *Drosophila* genome (*Morrison et al., 2000*). We used the GAL4-UAS system to drive available *UAS-RNAi* targeting 274 of the predicted 337 genes in the *Drosophila* kinome and phosphatome. *UAS-RNAi* were expressed in both neurons and muscle, and PHP was assessed at the third instar NMJ by direct measurement of synaptic transmission using intracellular recordings (*Figure 1A*). The rapid expression of PHP was assessed by application of sub-blocking concentrations of the glutamate receptor antagonist philanthotoxin-433 (PhTX; 15 μM), which causes a decrease in the amplitude of miniature excitatory postsynaptic potential amplitudes (mEPSPs) (*Frank et al., 2006*, *Figure 1A*). Excitatory postsynaptic potential amplitudes (EPSPs) initially decrease. However, within ten minutes, homeostatic signaling is engaged and drives an increase in presynaptic release (quantal content, calculated as EPSP/mEPSP). Increased presynaptic release precisely counteracts the decrease in receptor sensitivity and returns EPSP amplitudes back to the synaptic set point, baseline values. (*Figure 1A*; see *Frank et al., 2006*).

First, we generated a large data set to define the baseline parameters of PHP and quantify any effect caused by the heterozygous GAL4 lines used in our screen. We assessed PHP in heterozygous GAL4 lines (*GAL4/+*) crossed to a wild-type genetic background by recording mEPSP and EPSP amplitudes in the absence (baseline) or presence of PhTX. The data are plotted in *Figure 1B and C*, with each point representing average data from an individual NMJ recording. In *Figure 1B*, note that mEPSP amplitudes, across all recordings, decreased by 51.4% (see arrow, X-axis; p<0.001), while the average EPSP amplitude decreased by only 9.0% (p<0.001). When mEPSP amplitudes are plotted against quantal content, a significant correlation emerges ($R^2$ = 0.80; *Figure 1C*). This function (equation) defines the process of PHP whereby the magnitude of mEPSP decrease is offset by a corresponding increase in presynaptic release (quantal content). By plotting individual data, we can establish an interval that contains 95% of all data points within this control data set (blue lines). Average values and sample sizes can be found in *Supplementary file 1*, inclusive of data in all subsequent figures.

In our screen, we quantified average mEPSP, EPSP, quantal content, muscle input resistance and muscle resting membrane potential for each combination of *UAS-RNAi* and *GAL4* driver. The screen data are plotted (*Figure 1D–E*), with each point representing the average of multiple NMJ recordings for an individual genotype. The majority of individual data points shown on the graphs (94%) represent averages of more than three individual recordings, with sample sizes ranging from 2 to 14 recordings per genotype, totaling 1150 NMJ recordings. Knockdown genotypes (*UAS-RNAi/GAL4*) were tested in the presence of PhTX (yellow circles). A subset of genotypes was also tested in the absence of PhTX (black circles, inclusive of ~100 muscle recordings). The average mEPSP amplitude across all genotypes decreased by 51.6% (arrow X-axis; p<0.001) and the average EPSP amplitude decreased by 11.2% (p<0.001). The similarity with the control data (*Figure 1B*) is indicative of robust homeostatic compensation when the majority of the kinases and phosphatases were knocked down in our screen. When mEPSP amplitudes are plotted against quantal content, the majority of data reside within the confidence interval created for our control data set (*Figure 1E*, blue lines). Two criteria were used to select candidate PHP genes: 1) those that reside below a cutoff of −2 standard deviations from the average EPSP amplitude of the control genotypes and 2) those that reside outside the blue lines defining 95% of control data (superimposed on the screen data) (*Figure 1E*, blue lines). Finally, our data demonstrate that there is no correlation observed between quantal content and either resting membrane potential or muscle input resistance (*Figure 1—figure supplement 1*).

## Identification of class II PI3K as a homeostatic plasticity gene

We re-screened mEPSP and EPSP amplitudes in the presence and absence of PhTX, and identified 5 RNAi lines as verified hits from our screen (*Table 1*). These five lines target the PI3-Kinase *Pi3K68D*, the class III PI3-Kinase *Vps34,* the tyrosine-like kinase *cdi*, a putative kinase encoded by *CG8726*, and *CamKII*. In addition, we identified additional candidate plasticity genes with an apparent block of PHP (not yet independently verified) including PI4-Kinase (*PI4KIIIα*) and CamKK (unpublished data KJF and GWD). The discovery of *Pi3K68D* as well as *Vps34* and, potentially *PI4KIIIα,* from a screen of more than 250 kinases and phosphatases, strongly implicates lipid kinase signaling in the homeostatic control of presynaptic neurotransmitter release.

*Pi3K68D* was one of the most robust hits from the screen with a very large change in mEPSP amplitude and a similarly large decrease in EPSP amplitude (see red line on 1D), indicative of a complete block of PHP (no difference in QC, p=0.17, *Figure 1D,E*). We chose to initially focus on the function of *Pi3K68D*, in part due the robustness of the phenotype and in part because very little is known about the function of class II PI3Ks within the nervous system of any organism. As such, we have the opportunity to expand the general knowledge of lipid kinase signaling pathways and define new mechanisms underlying PHP.

## *Pi3K68D* encodes a class II PI3K that is necessary for PHP

*Pi3K68D* encodes a class II PI3K with homology to the three Class II PI3Ks encoded in the mammalian genome [*Figure 2A*; homology to PIK3C2A is 31% identical and 48% similar (*Sievers et al., 2011*). In order to pursue a formal genetic analysis of *Pi3K68D* we examined two existing transposon insertion mutations in the *Pi3K68D* locus (*Pi3K68D^{GS}* residing in the 5' UTR and *Pi3K68D^{MB}* residing in an intron; *Figure 2A*). Since neither transposon resides in coding sequence, we also used the CRISPR-Cas9 system (*Kondo and Ueda, 2013*) to generate a new mutation *Pi3K68D^{AH1}*. This mutation is a small insertion/deletion mutation resulting in a premature stop codon at amino acid 1440, prior to the kinase domain. Presynaptic homeostatic plasticity was fully blocked in all three *Pi3K68D* mutations (*Figure 2B,C*). Unless noted, all following experiments are done with the *Pi3K68D^{AH1}* CRISPR mutant. These data confirm the results of our RNAi-based screen, identifying *Pi3K68D* as an essential gene for PHP.

To further define the extent to which loss of *Pi3K68D* affects baseline transmission and PHP, we plotted the data for each individual NMJ recording, comparing mEPSP amplitude and quantal content (*Figure 2D*). Sample size for baseline recordings from *Pi3K68D* in the absence of PhTX includes 75 individual recordings, and 52 individual recordings in the presence of PhTX. At wild-type synapses, as shown in *Figure 1*, quantal content increased homeostatically as average mEPSP amplitude decreased. Again, the data are fit with an exponential function indicative of homeostatic plasticity sustaining set point postsynaptic excitation ($R^2 = 0.62$; solid line *Figure 2D*, left). By contrast,

**Table 1.** Selected Hits from Screen

| Gene name | Driver | N | PhTX | RMP | Rin | mEPSP amplitude (mV) | EPSP amplitude (mV) | Quantal content |
|---|---|---|---|---|---|---|---|---|
| Control | *Driver/+* | 94 | - | −67.0 ± 0.5 | 8.7 ± 0.2 | 0.93 ± 0.02 | 34.6 ± 0.6 | 39.3 ± 1.0 |
| Control | *Driver/+* | 141 | + | −66.8 ± 0.5 | 8.7 ± 0.2 | 0.45 ± 0.01 | 31.5 ± 0.5 | 76.3 ± 1.8 |
| *Pi3K68D* | *Sca-GAL4/+; BG57-GAL4/+* | 9 | - | −70.0 ± 1.9 | 6.6 ± 0.35 | 1.23 ± 0.07 | 33.6 ± 2.6 | 28.0 ± 2.6 |
| *Pi3K68D* | *Sca-GAL4/+; BG57-GAL4/+* | 11 | + | −66.0 ± 1.3 | 7.1 ± 1.7 | 0.50 ± 0.09 | 17.2 ± 2.8 | 41.1 ± 2.3 |
| *Vps34* | *Sca-GAL4/+; BG57-GAL4/+* | 8 | - | −63.9 ± 1.35 | 5.8 ± 0.6 | 0.98 ± 0.05 | 32.4 ± 0.5 | 33.7 ± 4.2 |
| *Vps34* | *Sca-GAL4/+; BG57-GAL4/+* | 13 | + | −67.2 ± 1.7 | 6.1 ± 1.5 | 0.48 ± 0.04 | 20.9 ± 2.1 | 45.1 ± 4.4 |
| *cdi* | *OK371-GAL4/+; BG57-GAL4/+* | 7 | - | −70.2 ± 1.3 | 8.3 ± 0.4 | 0.93 ± 0.05 | 28.2 ± 1.5 | 30.7 ± 2.2 |
| *cdi* | *OK371-GAL4/+; BG57-GAL4/+* | 12 | + | 64.9 ± 1.6 | 8.5 ± 0.5 | 0.41 ± 0.03 | 13.6 ± 1.6 | 23.9 ± 4.9 |
| *cg8726* | *OK371-GAL4/+; BG57-GAL4/+* | 12 | - | −73.5 ± 1.9 | 8.9 ± 0.4 | 1.1 ± 0.11 | 30.9 ± 2.4 | 28.9 ± 2.4 |
| *cg8726* | *OK371-GAL4/+; BG57-GAL4/+* | 8 | + | −66.7 ± 1.7 | 10.0 ± 0.4 | 0.45 ± 0.03 | 17.1 ± 1.8 | 39.3 ± 4.2 |
| *CamKII* | *OK371-GAL4/+; BG57-GAL4/+* | 7 | - | −64.1 ± 1.3 | 7.8 ± 0.5 | 0.83 ± 0.03 | 24.9 ± 2.7 | 30.5 ± 3.7 |
| *CamKII* | *OK371-GAL4/+; BG57-GAL4/+* | 7 | + | −61.0 ± 2.3 | 6.7 ± 1.6 | 0.41 ± 0.05 | 17.7 ± 1.4 | 46.0 ± 5.9 |

DOI: https://doi.org/10.7554/eLife.31535.004

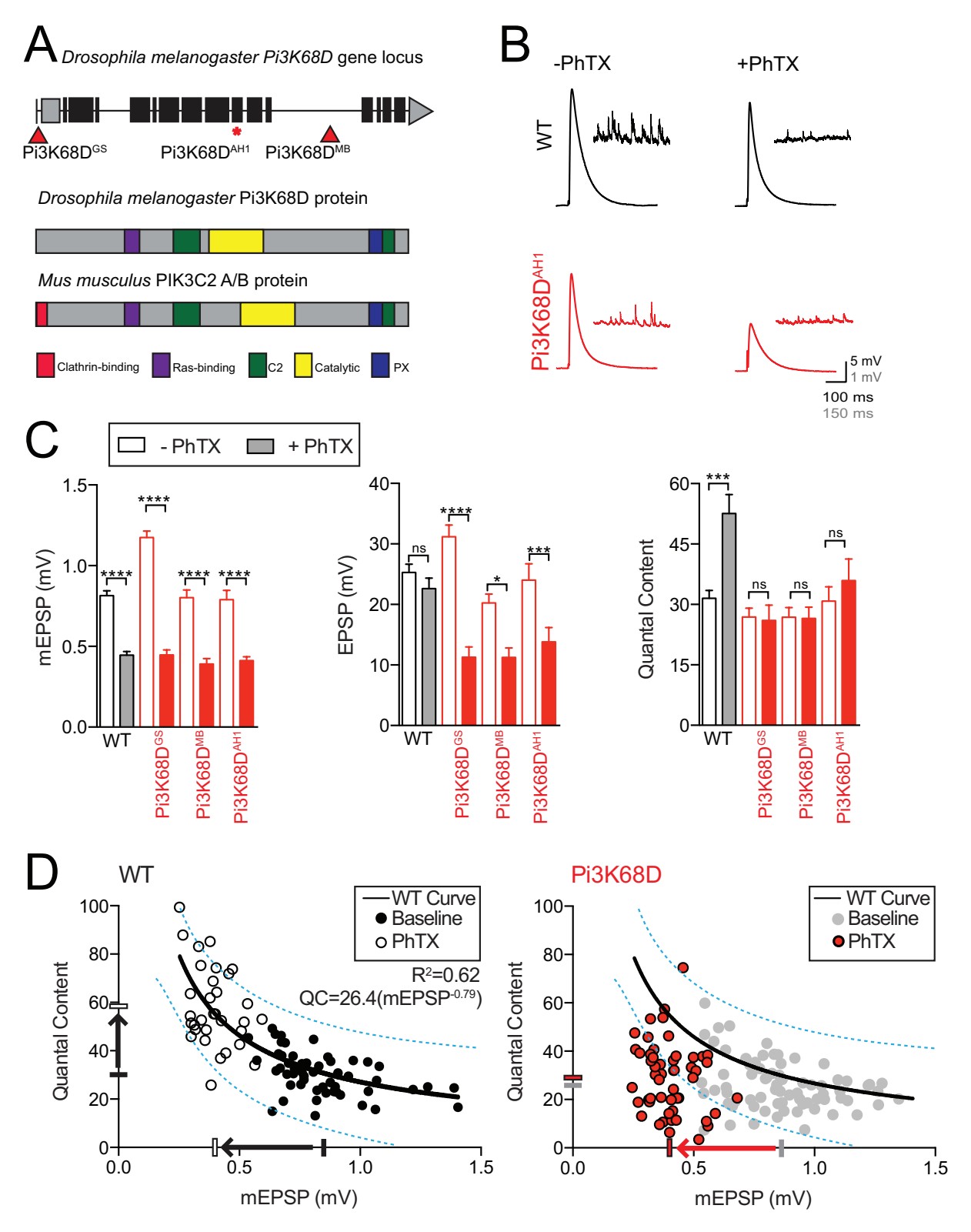

**Figure 2.** Mutations in *Pi3K68D* block the rapid expression of presynaptic homeostatic plasticity. (**A**) Top: Schematic of the *Pi3K68D* gene locus with mutations indicated. Below: Pi3K68D protein domains are indicated and compared to mouse class II PI3K proteins. (**B**) Representative EPSP and spontaneous mEPSP in wild-type (black) and mutant *Pi3K68D* (red), in the absence or presence of PhTX as indicated. Recordings made at 0.3 mM $[Ca^{2+}]_e$. (**C**) Average mEPSP amplitude, EPSP amplitude, and presynaptic release (quantal content) in *WT*, *Pi3K68D^GS*, *Pi3K68D^MB*, and *Pi3K68D^AH1*.
*Figure 2 continued on next page*

*Figure 2 continued*
Unfilled bars are in the absence of PhTX. Filled bars are in the presence of PhTX. Mean ± SEM; ns not significant, *p<0.05, **p<0.01, ***p<0.001, and
****p<0.0001; Student's *t*-test. (**D**) Each point represents average data from an individual NMJ recording. For WT, recordings in absence of PhTX are
filled black circles while those with PhTX are unfilled black circles. For *Pi3K68D,* recordings in absence of PhTX are grey circles while those with PhTX
are red circles. The black or grey filled hash marks on the X and Y axes represent the average mEPSP amplitude and QC, respectively, without PhTX.
The white and red hash marks on the X and Y axes represent average mEPSP amplitude and QC, respectively, with PhTX. The black line in the WT
graph is a curve fit to this control data. The same wild-type curve-fit is overlaid on the *Pi3K68D* data for purposes of comparison. Dotted blue lies
encompass 95% of wild-type data points. These same lines from wild-type are superimposed on the *Pi3K68D* graph, at right.
DOI: https://doi.org/10.7554/eLife.31535.005

*Pi3K68D* mutant synapses do not adhere to the wild-type homeostatic function (*Figure 2D*, right, red). The fit from the wild-type data set, including the interval that contains 95% of the wild-type data (blue lines), is superimposed on the *Pi3K68D* data. A majority of all *Pi3K68D* data points lie outside our 95% control data interval (defined above). There is no statistically significant change in the average QC in the presence of PhTX despite a ~ 5 fold decrease in mEPSP amplitude in *Pi3K68D,* implying a failure of PHP (*Figure 2*). We also note that there is a small but significant decrease in baseline QC (minus PhTX) in the *Pi3K68D* mutant background (WT QC = 30.3 ± 1.2 and *Pi3K68D* QC = 26.1 ± 1.2; p=0.018; see below for further analysis and discussion of baseline transmission).

## *Pi3K68D* is necessary for the long-term maintenance of homeostatic potentiation

PHP can be induced by genetic deletion of the muscle-specific glutamate receptor subunit *GluRIIA* (*Petersen et al., 1997*). This is an independent method to induce PHP, and it has been interpreted to reflect the long-term maintenance of PHP throughout lifespan (*Mahoney et al., 2014*; *Marie et al., 2010*). Genes involved in both the rapid expression and the long-term maintenance of PHP can be considered to be 'core' genes necessary for PHP. We found that *GluRIIA; Pi3K68D^{AH1}* double mutant animals have decreased EPSP amplitude due to a failure to homeostatically increase quantal content (*Figure 3A,B*). Collectively these data show *Pi3K68D* is necessary for both the rapid expression and long-term maintenance of PHP and, as such, can be considered part of the core homeostatic machinery necessary for PHP.

## *Pi3K68D* mutants have normal morphology and glutamate receptor abundance

We quantified various measures of synapse development and morphology to determine whether the disruption of homeostatic plasticity in *Pi3K68D* mutants might be secondary to changes in synaptic structure. There was no difference in bouton number or in active zone number (as quantified by Bruchpilot, BRP, a key component of the presynaptic active zone) comparing wild-type and *Pi3K68D* mutants (*Figure 3—figure supplement 1*). We found no significant difference between wild-type and *Pi3K68D* mutant larva in GluRIIA receptor subunit intensity (*Figure 3—figure supplement 1*). There was a small but significant increase in GluRIIB intensity (16% increase compared to wild-type, p=0.01, data not shown). We found no difference in average intensity of two additional synaptic markers including Clathrin Light Chain (CLC) and cysteine string protein (CSP, *Figure 3—figure supplement 1*). There was a small, significant increase in synaptic anti-synaptotagmin-1 intensity (27% increase, p=0.006. *Figure 3—figure supplement 1*). In conclusion, we find no evidence for a substantial decrease in key presynaptic proteins, postsynaptic neurotransmitter receptors, or bouton numbers in the *Pi3K68D* mutant.

## *Pi3K68D* is required postsynaptically for PHP

To determine where *Pi3K68D* functions during PHP, pre- versus postsynaptically, we performed tissue-specific rescue experiments by expressing *UAS-Pi3K68D-GFP* (*Velichkova et al., 2010*) in the *Pi3K68D* mutant background. First, overexpression of *UAS-Pi3K68D-GFP* in motoneurons had no significant effect on baseline transmission and failed to rescue PHP when expressed in the *Pi3K68D* mutant background (*Figure 4A*). Next, muscle specific over-expression of *UAS-Pi3K68D-GFP* using *MHC-GAL4* impaired muscle health and diminished baseline transmission (*Figure 4B*). Muscle-specific expression of *UAS-Pi3K68D-GFP* with the *BG57-GAL4* driver was not viable (*MacDougall et al.,*

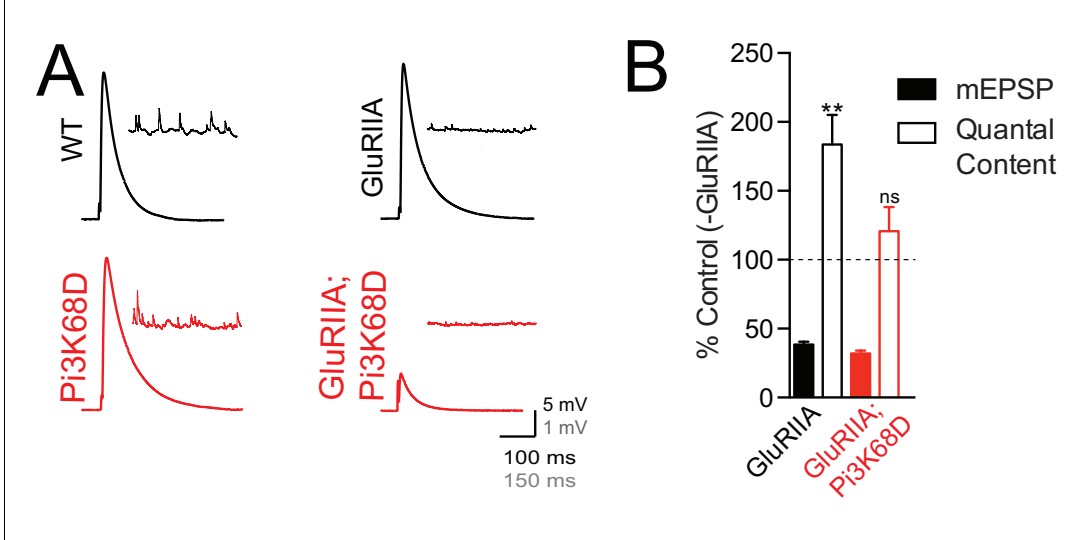

**Figure 3.** Loss of *Pi3K68D* blocks the sustained expression of presynaptic homeostatic plasticity. (**A**) Representative EPSP and spontaneous mEPSP in wild-type (black) and *Pi3K68D^{AH1}* (red), at baseline or in the presence of the *GluRIIA* mutation as indicated. Recordings made at 0.3 mM [Ca²⁺]ₑ. (**B**) Average percent change in mEPSP amplitude (filled bars) and quantal content (open bars) in loss of *GluRIIA* compared to baseline for indicated genotypes. Statistics comparing baseline and PhTX conditions for quantal content for indicated genotypes. Mean ± SEM; ns not significant, **p<0.01; Student's *t*-test.

DOI: https://doi.org/10.7554/eLife.31535.006

The following figure supplement is available for figure 3:

**Figure supplement 1.** Normal NMJ anatomy.

DOI: https://doi.org/10.7554/eLife.31535.007

*2004*). However, over-expression of *UAS-Pi3K68D-GFP* with *BG57-GAL4* in the *Pi3K68D* mutant background was viable and fully rescued PHP (*Figure 4B*). Muscle recordings from these rescue animals revealed impaired muscle resting membrane potentials (RMP) due to *Pi3K68D* overexpression (RMP = −60.1 ± 1.3 mV without PhTX and −59.3 ± 2.4 mV with PhTX compared to the *Pi3K68D* mutant: −68.2 ± 1.3 mV without PhTX and −68.0 ± 1.8 mV with PhTX). We find an associated decrease in mEPSP amplitude, likely due to diminished driving force (*Figure 4B*). None-the-less, postsynaptic expression of *Pi3K68D* fully restored PHP. Thus, *Pi3K68D* is necessary postsynaptically for PHP.

We next sought to determine whether Pi3K68D kinase activity is required for PHP. A previously published kinase dead transgene for *Pi3K68D* is no longer available (*MacDougall et al., 2004*). Therefore, we generated a new *UAS-kinase-dead Pi3K68D* by creating a small deletion (21 amino acids) within the kinase domain (*Figure 4C*), termed *UAS-KDΔ21-Pi3K68D*. To confirm that this transgene inhibits kinase activity when expressed in a wild-type genetic background, we expressed *UAS-KDΔ21-Pi3K68D* in the scutellar bristle lineage using *ptc-GAL4*. This produced supernumerary scutellar bristles, phenocopying the effects of the previously published kinase dead transgene (data not shown; *MacDougall et al., 2004*). When we over-expressed *UAS-KDΔ21-Pi3K68D* in muscle, PHP was blocked (*Figure 4C and D*). A C-terminal m-Cherry tag allowed us to determine that the transgene was expressed in muscle (data not shown). In addition, muscle overexpression of *UAS-KDΔ21-Pi3K68D* (*BG57-GAL4*) had no adverse effect on muscle health, unlike over-expression of wild-type *Pi3K68D* with *BG57-GAL4,* which causes lethality (see above). From these data, we conclude that the kinase activity of *Pi3K68D* is necessary for PHP. These data further confirm that *Pi3K68D* is necessary, in muscle, for PHP.

The N-terminal amino acids of mammalian PIK3C2A and PIK3C2B have a regulatory function, binding Clathrin and regulating the activity of the kinase (*Gaidarov et al., 2005*; *Wheeler and Domin, 2006*). Therefore, we sought to test whether disrupting this regulatory function would alter the rapid expression of PHP. We generated an N-terminal *Pi3K68D* deletion, removing the

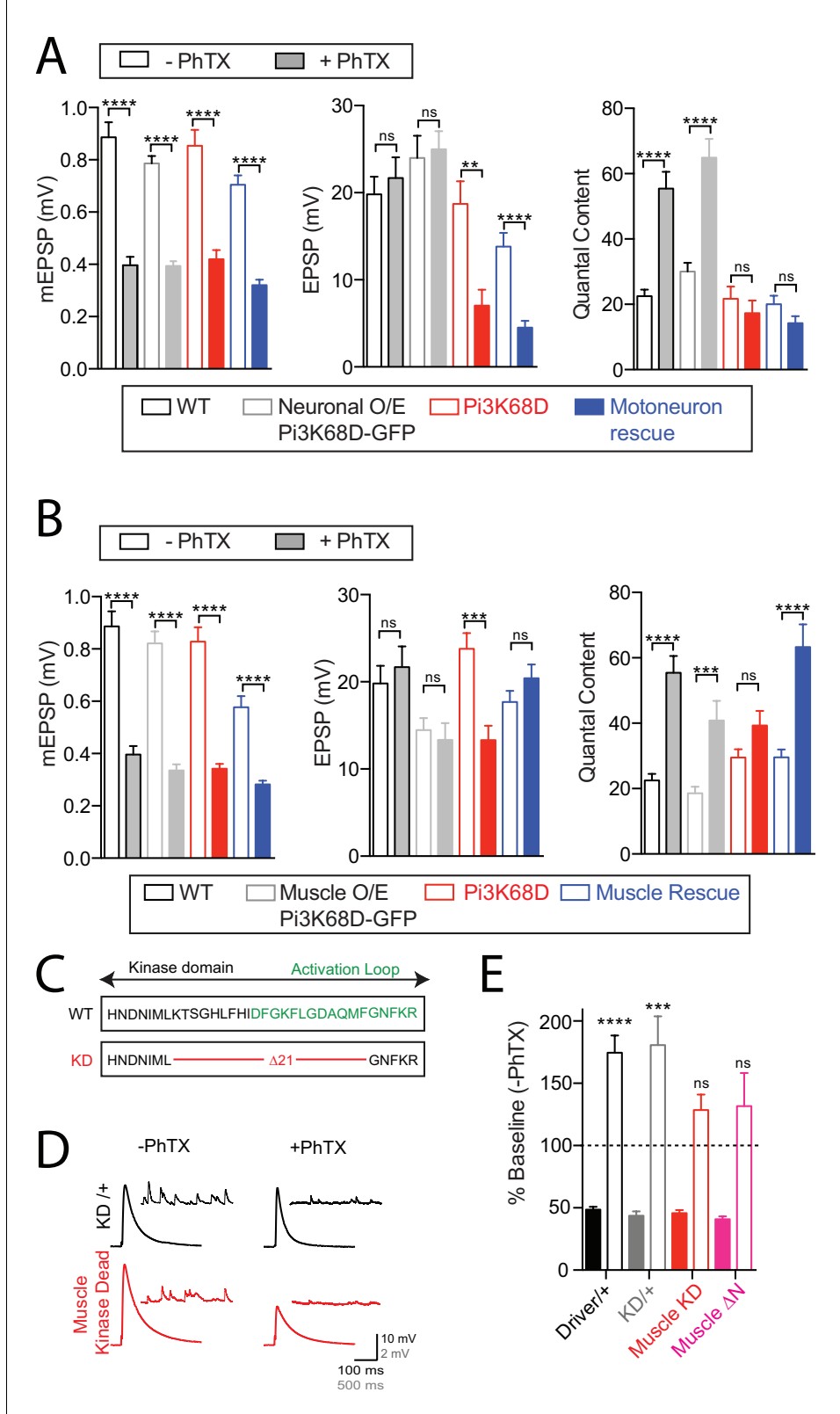

**Figure 4.** Postsynaptic *Pi3K68D* is necessary for presynaptic homeostatic plasticity. (**A**) Average data for measures as indicated, for conditions, as indicated. Genotypes tested: *UAS-Pi3K68D-GFP* driven by *OK371-GAL4*; (light grey), *Pi3K68D^{AH1}* (red), and *Pi3K68D^{AH1}* with *UAS-Pi3K68D-GFP* driven by *OK371-GAL4*; (blue). Recordings made at 0.3 mM [Ca$^{2+}$]$_e$. (**B**) Data as in (**A**). Genotypes tested WT (data re-plotted from A for direct comparison; black), *UAS-Pi3K68D-GFP* driven by *MHC-GAL4* (grey), *Pi3K68D^{AH1}* (red), and *Pi3K68D^{AH1}* with *UAS-Pi3K68D-GFP* driven by *BG57-GAL4* (blue).

*Figure 4 continued on next page*

Figure 4 continued

Recordings made at 0.3 mM [Ca$^{2+}$]$_e$. (C) Schematic, beginning at Pi3K68D amino acid 1441, of the mutation used to generate a kinase-dead mutant transgene. (D) Representative EPSP and spontaneous mEPSP in *UAS-Pi3K68D-KD/+* (black) and *UAS-Pi3K68D-KD* driven by *BG57-GAL4* (red), in the absence or presence of PhTX as indicated. Recordings at 0.3 mM Ca$^{2+}$. (E) Average percent change in mEPSP amplitude (filled bars) and quantal content (open bars) in PhTX compared to baseline in *BG57-GAL4/+* (black), *UAS-Pi3K68D-KD/+* (grey), *UAS-Pi3K68D-KD* driven by *BG57-GAL4* (red), and *UAS-Pi3K68D-ΔN* driven by *BG57-GAL4,* (magenta). Muscle driver is BG57-GAL4. Mean ±SEM; ns not significant, *p<0.05, **p<0.01, ***p<0.001, ****p<0.0001; Student's *t*-test.

DOI: https://doi.org/10.7554/eLife.31535.008

orthologous first 363 amino acids, termed *UAS-Pi3K68D-ΔN*. When we over-expressed *UAS-Pi3K68D-ΔN* in muscle, PHP was blocked (*Figure 4E*). When Pi3K68D-ΔN protein localization was followed using an N-terminal mCherry tag, it was apparent that the transgene is more diffuse compared to the punctate distribution of a wild-type *UAS-Pi3K68D-GFP* transgene (see below and data not shown). We conclude that either proper kinase localization or regulation, potentially via Clathrin binding, is essential for robust homeostatic signaling. Collectively, these results demonstrate a required postsynaptic function of *Pi3K68D* for presynaptic homeostatic plasticity.

## *Pi3K68D* genetically interacts with previously characterized PHP genes

The genes encoding Rab3 Interacting molecule (RIM; *Müller et al., 2012*), a central player in the presynaptic cytomatrix, and Multiplexin (DMP; *Wang et al., 2014*), a component of the extracellular matrix and precursor to Endostatin, have been shown to be required for PHP. Since homozygous mutations in each gene block PHP, we cannot perform standard double mutant genetic epistasis experiments. However, it has been possible to test double heterozygous mutant combinations for a block in PHP, thereby implicating genes in a common process, even if it is not possible to order them in single signaling pathway (*Frank et al., 2009*; *Harris et al., 2015*; *Wang et al., 2014*). Here we demonstrate that heterozygous mutations for *Pi3K68D/+* as well as *rim/+* and *dmp/+* all express normal PHP (*Figure 5A,B*). However, trans-heterozygous animals for *Pi3K68D* and *rim* show a complete block in PHP (*Figure 5A,B*). In addition, trans-heterozygous animals for *Pi3K68D* and *dmp* show a statistically significant suppression of homeostatic plasticity (*Figure 5A,B*). These data are consistent with *Pi3K68D* functioning within a homeostatic signaling system inclusive of previously identified PHP genes. Since RIM is a presynaptic protein and Pi3K68D acts postsynaptically, these data further imply that these genes participate, at some level, in a trans-synaptic signaling system necessary for PHP.

Multiplexin is a extracellular matrix protein, and its cleavage product Endostatin is hypothesized to act as a retrograde signal during PHP (*Wang et al., 2014*). Expression of *UAS-endostatin* in neurons or muscles rescues *dmp* mutants during PHP, consistent with a requirement for Endostatin being secreted into the synaptic cleft. Because *dmp* and *Pi3K68D* showed a genetic interaction as trans-heterozygotes (strong suppression of PHP), we hypothesized that Pi3K68D may be necessary for the secretion of Endostatin. In this model, expression of *UAS-endostatin* would rescue PHP in *Pi3K68D* mutants. We expressed *UAS-endostatin* in the muscle of *Pi3K68D* mutants and found that homeostatic plasticity was still blocked (*Figure 5—figure supplement 1*). In this condition, we confirmed that *Pi3K68D* mutants do not inhibit secretion of Endostatin-GFP by labeling surface GFP under non-cell permeabilizing conditions (*Figure 5—figure supplement 1*). As a control for exclusive labeling of secreted protein, we demonstrate that a highly over-expressed cytoplasmic GFP shows not labeling under non-cell permeabilizing conditions (see S6K-GFP, an intracellular protein tagged with GFP). Therefore, we conclude that secreted Endostatin is not sufficient to rescue the defect in PHP in *Pi3K68D* mutants. As such, it is unlikely that impaired PHP is due to a failure to release Multiplexin or proteolytically process Muliplexin into Endostatin within the synaptic cleft.

## Loss of Pi3K68D disrupts a postsynaptic PI(3)P-Dependent Endosomal System

The class II PI3 kinases (PI3K-cII) remain poorly characterized, particularly when compared to other components of lipid and endosomal signaling systems. It has been reported in other cell biological systems that PI3K-cII binds to Clathrin and is activated, in part, through this interaction (*Domin et al., 2000*; *Gaidarov et al., 2005*; *Wheeler and Domin, 2006*). In addition, PI3K-cII drives

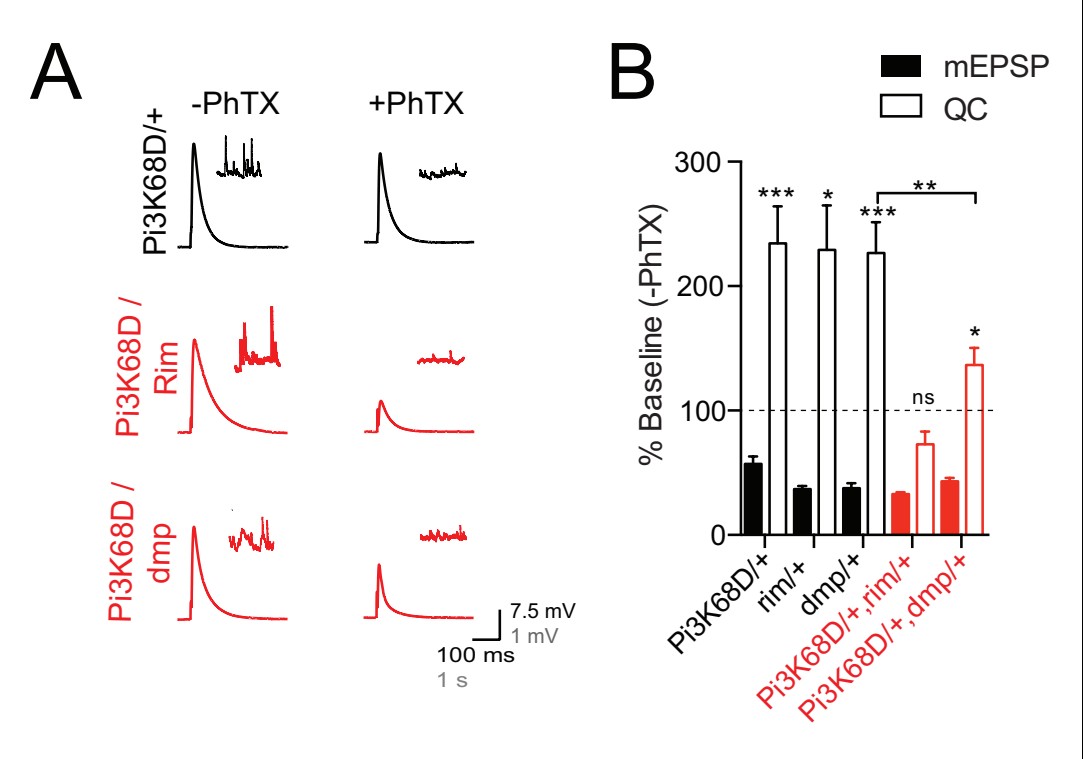

**Figure 5.** *Pi3K68D* interacts genetically with *rim* and *dmp*. (**A**) Representative traces for EPSP and mEPSP in heterozygous controls (*Pi3K68D*[AH1]/+; black) and double heterozygous mutants *Pi3K68D*[AH1]/*rim*[103] (red) and *Pi3K68D*[AH1]/*dmp*[f0] (red) in the absence or presence of PhTX, as indicated. Recordings made at 0.3 mM [Ca$^{2+}$]$_e$. (**B**) Average percent change in mEPSP amplitude (filled bars) and quantal content (open bars) in PhTX compared to baseline in heterozygous controls: *Pi3K68D*[AH1]/+ (black), *rim*[103]/+ (black), and *dmp*[f0]/+ (black) and double heterozygous mutants: *Pi3K68D*[AH1]/*rim*[103] (red) and *Pi3K68D*[AH1]/*dmp*[f0] (red).

DOI: https://doi.org/10.7554/eLife.31535.009

The following figure supplement is available for figure 5:

**Figure supplement 1.** Endostatin does not rescue *Pi3K68D*.

DOI: https://doi.org/10.7554/eLife.31535.010

the generation of the lipid PI(3)P (*MacDougall et al., 2004*), which defines an early endosomal membrane system (*Posor et al., 2015*). A plausible scenario for the localization and function of PI3K-cII in the secretory pathway is diagrammed (*Figure 6A,B*). This model serves as a guide to our experiments (below).

To visualize the subcellular localization of Pi3K68D, we expressed *UAS-Pi3K68D-GFP* in muscle, co-stained with anti-Clathrin Light Chain (CLC) (*Heerssen et al., 2008*) and imaged the preparation using super-resolution structured illumination microscopy (*Pielage et al., 2008*). We demonstrate that Pi3K68D-GFP forms endosomal like structures that precisely co-localize with CLC throughout muscle, concentrating near the muscle surface (*Figure 6C*). The highly regular distribution of Pi3K68D puncta is reminiscent of the distributed Golgi system in skeletal muscle, observed in mammals (*Ralston et al., 2001*) and *Drosophila* (*Johnson et al., 2015*). Therefore, we attained YFP or GFP-tagged markers of the medial and trans-Golgi (*Ye et al., 2007*). We tested for co-localization of our Golgi markers with CLC antibody staining, since anti-CLC is co-localized with Pi3K68D-GFP. We find that CLC and trans-Golgi-YFP (GalT-YFP) reside in closely associated vesicular compartments throughout muscle (*Figure 6D*). This finding is confirmed by use of a second medial Golgi-GFP marker (*UAS-ManII-GFP*) (*Figure 6—figure supplement 1*). We conclude that Pi3K68D is present on a Golgi-derived, Clathrin-positive membrane compartment, consistent with prior work in other systems (*Domin et al., 2000*; *Wheeler and Domin, 2006*).

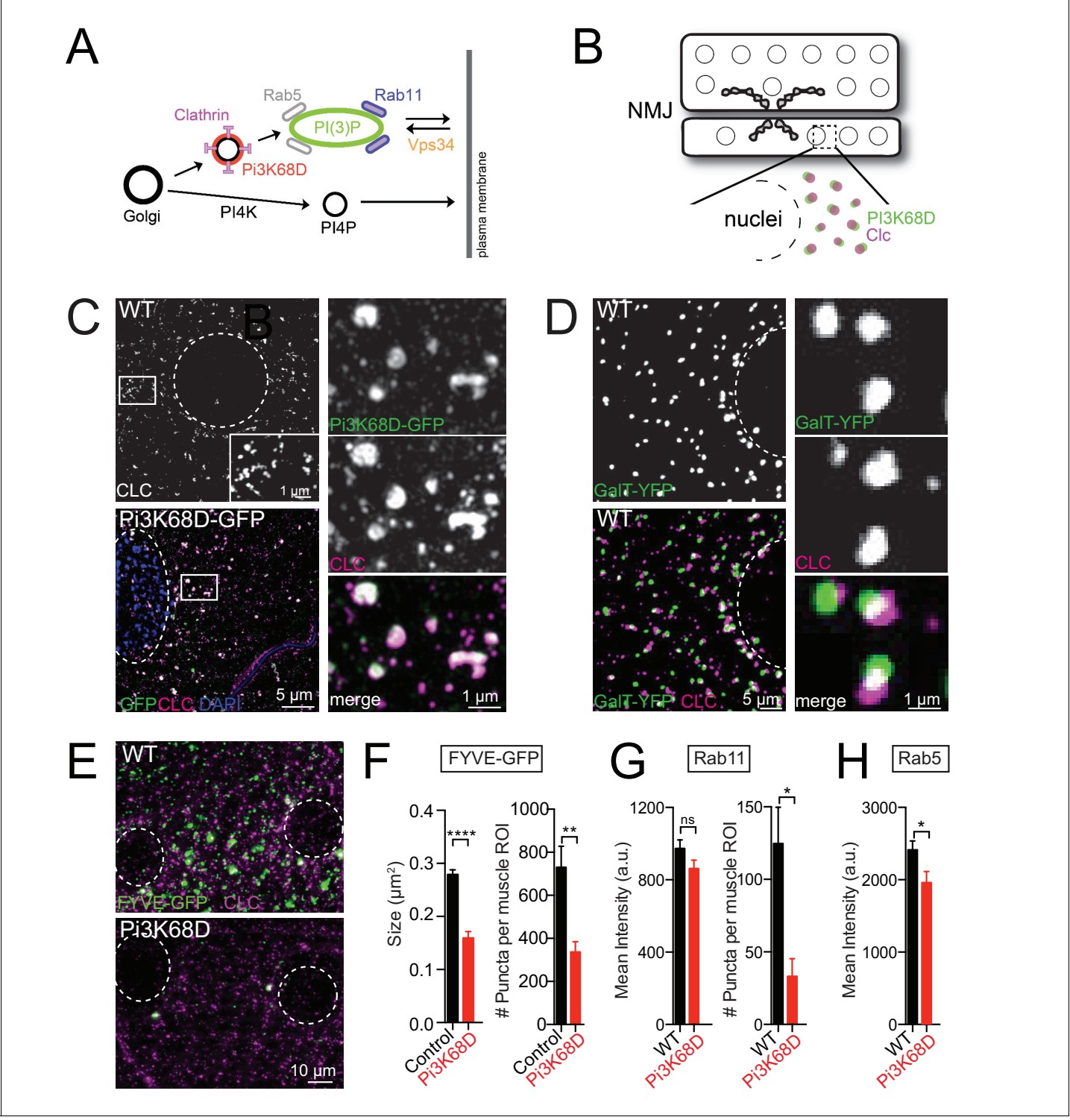

**Figure 6.** The role of *Pi3K68D* in endosomal and trans-Golgi signaling. (**A**) A model of the Golgi and endosomal system illustrating proteins explored in this and subsequent figures. (**B**) A schematic of the NMJ, muscles 6 and 7. Large circles depict muscle nuclei. Synaptic boutons are outlined in black and filled gray. Dotted box indicates a region of interest (ROI). The ROI, expanded below, reveals the edge of a nucleus, and puncta defined by Pi3K68D (green) and CLC (magenta). This schematizes images presented in C-E. (**C**) Muscle ROI in WT and animals expressing *UAS-Pi3K68D-GFP* in muscle (*MHC-GAL4*). Staining as indicated; GFP (green) and CLC (magenta or white in WT image). Nuclei are outlined with a white dotted line. Insets at right. (**D**) ROI in muscle expressing *GalT-YFP* (trans-Golgi). Staining as indicated. Select puncta shown at higher magnification (right). (**E**) Muscle ROI in WT and *Pi3K68D* [AH1] expressing *UAS-2xFYVE-GFP*. Staining as indicated. Nuclei as in (**C**). (**F**) FYVE-GFP puncta area and number per muscle ROI. Black

*Figure 6 continued on next page*

*Figure 6 continued*

is control (*FYVE-GFP/+;BG57-GAL4/+*). Red is *FYVE-GFP/+;BG57-GAL4, Pi3K68D* [AH1]/*Pi3K68D* [AH1]. Number of synapses: WT N = 10, *Pi3K68D* N = 9. (**G**) Rab11 mean intensity and puncta number per muscle ROI. WT (black) versus *Pi3K68D* [AH1] (red). Number of synapses: WT N = 10, *Pi3K68D* N = 7. (**H**) Mean intensity per ROI of Rab5. WT (black) versus *Pi3K68D* [AH1] (red). Number of synapses: WT N = 12, *Pi3K68D* N = 11. Mean ±SEM; ns not significant, *p<0.05, **p<0.01, ***p<0.001, and ****p<0.0001; Student's *t*-test.

DOI: https://doi.org/10.7554/eLife.31535.011

The following figure supplements are available for figure 6:

**Figure supplement 1.** Distribution of ManII-GFP in muscle.

DOI: https://doi.org/10.7554/eLife.31535.012

**Figure supplement 2.** Images and quantification of endosomal antigens.

DOI: https://doi.org/10.7554/eLife.31535.013

We next determined whether Pi3K68D is responsible for the generation of PI(3)P by quantifying the distribution of a *UAS-2xFYVE-GFP* transgene, encoding a FYVE-domain protein that directly binds to PI(3)P (*Hammond and Balla, 2015*). We found that *Pi3K68D* mutants have dramatically fewer, smaller FYVE-GFP puncta compared to wild-type larvae (*Figure 6E,F*). As a control, we quantified CLC levels in the same muscles and find that they are unaffected (*Figure 6—figure supplement 2*). Furthermore, FYVE-GFP puncta do not co-localize with CLC, consistent with the presence of Pi3K68D on an intermediate endosomal membrane that is required for the formation of PI(3)P-positive endosomes. Thus, Pi3K68D is required to generate a significant fraction of PI(3)P in postsynaptic muscle, a necessary step in the formation of early endosomes in other systems (*Posor et al., 2015*).

It has been established that PI(3)P production at the early endosome recruits effectors that sort proteins to late endosomes, autophagosomes, or recycling endosomes (*Posor et al., 2015*). The proteins Rab5, Rab7, and Rab11 mark early endosomes, late endosomes, and recycling endosomes respectively (*Grant and Donaldson, 2009*). We quantified endogenous protein levels of Rab5, Rab7, and Rab11 in the muscle of wild-type and *Pi3K68D* mutant larva. Rab5, 7, and 11 are all enriched at the synapse, but we cannot differentiate between pre-and postsynaptic proteins by antibody staining. Therefore, we focused our examinations in regions of interest in the muscle, adjacent to the synapse. We found that loss of *Pi3K68D* does not affect total muscle protein levels of Rab11, but there is a significant drop in the number of Rab11-positive puncta (124.7 in WT versus 33.14 in *Pi3K68D* mutant, p=0.01; Student's *t*-test, two tailed; *Figure 6G*). We also examined levels of Rab5 in the *Pi3K68D* mutant and found that mean intensity in the muscle ROI also decreased (*Figure 6H*). The Rab5-positive puncta could not be sufficiently resolved to quantify puncta number. Finally, we find that total Rab7 protein increased in the *Pi3K68D* mutant (quantified by mean intensity), but the number of Rab7-positive puncta did not change (*Figure 6—figure supplement 2*). These data are consistent with the depletion of PI(3)P in the *Pi3K68D* mutant background causing impaired Rab11 recruitment to endosome derived vesicles that recycle to and from the plasma membrane. If so, this membrane trafficking system could be a signaling platform required for the rapid expression of PHP.

## Postsynaptic Rab11 is necessary for PHP

Rab11-positive endosomes are often referred to as recycling endosomes. These endosomes receive cargo, sorted within the PI(3)P positive endosome, and recycle the cargo back to the plasma membrane, thereby controlling the steady state concentration of important signaling molecules at the plasma membrane (*Choy et al., 2014*; *Issman-Zecharya and Schuldiner, 2014*; *Zhang et al., 2011*). We specifically knocked down *Rab11* in muscle using a published *UAS-Rab11-RNAi* transgene (*Beckett et al., 2013*; *Dietzl et al., 2007*; *Xiong et al., 2012*) and a muscle-specific source of GAL4 (*BG57-GAL4*). We found that PHP was completely blocked (*Figure 7A,B*). It seems reasonable to conclude that the impaired Rab11 recruitment to early endosomes, and impaired generation/function of recycling endosomes, is responsible for the block of PHP in the *Pi3K68D* mutant.

## Postsynaptic Vps34 is necessary for PHP

In most biological systems, the formation of the recycling endosomal compartment requires the action of a class III PI3K (PI3K-cIII, *Figure 6A*), referred to here as Vps34 for consistency with the yeast nomenclature. The Vps34 complex has been studied extensively and is thought to be the

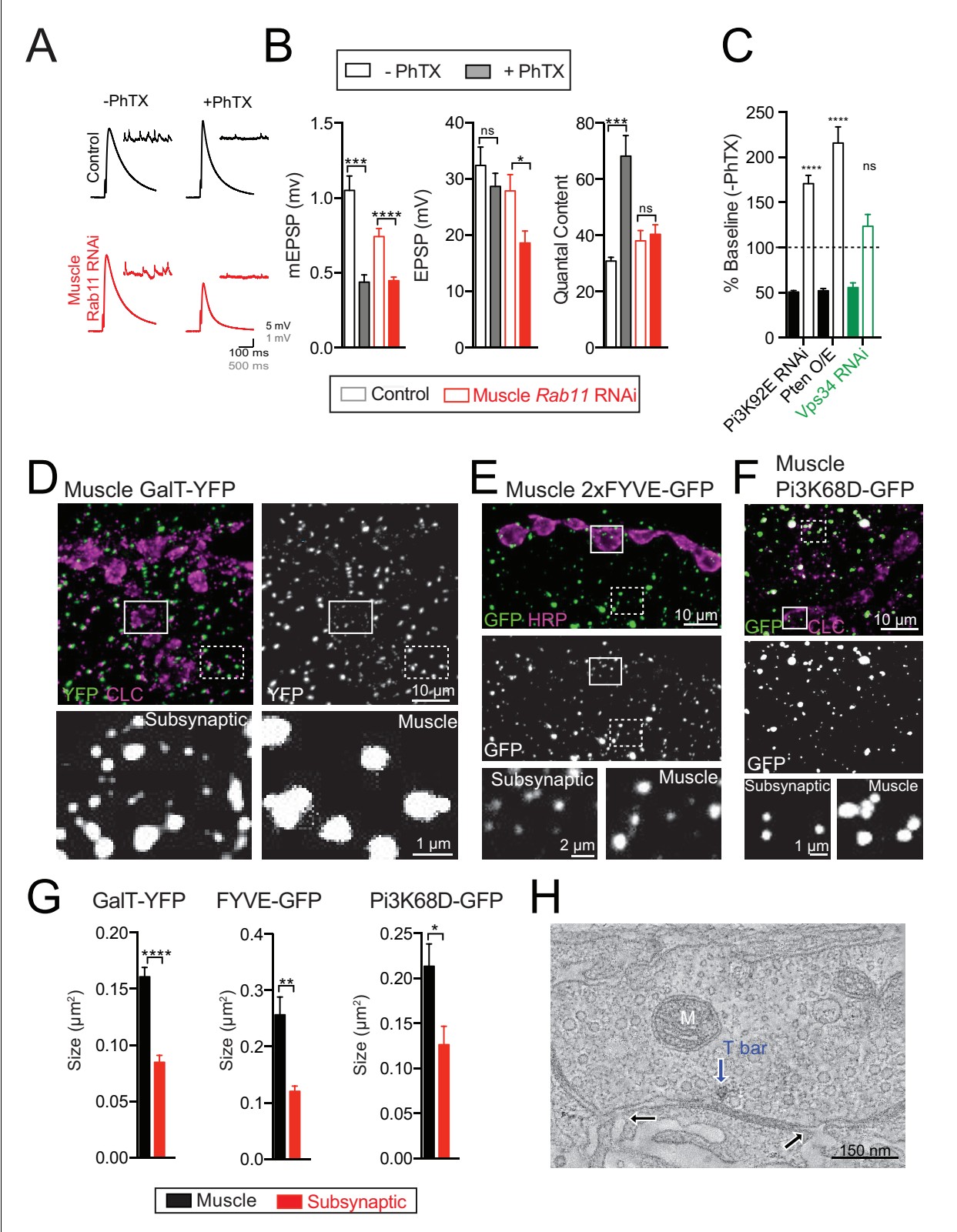

**Figure 7.** Sub-synaptic specialization of a trans-Golgi network. (**A**) Representative traces for controls (*BG57-GAL4/+*; black) and *UAS-Rab11-RNAi* driven in muscle (red; *BG57-GAL4*) in the absence or presence of PhTX, as indicated. Recordings at 0.3 mM [Ca$^{2+}$]$_e$. (**B**) Genotypes tested are control (*BG57-GAL4/+*; grey) and *UAS-Rab11-RNAi* (red; muscle-expression via *BG57-GAL4*). (**C**) Average percent change in mEPSP amplitude (filled bars) and quantal content (open bars) in PhTX compared to baseline. *UAS-Pi3K92E-RNAi* (black; muscle-expression via *MHC-GAL4*), *UAS-Pten* over-expression (black;

*Figure 7 continued on next page*

*Figure 7 continued*

muscle-expression via *BG57-GAL4*), and *UAS-Vps34-RNAi* (green; muscle-expression via *MHC-GAL4*). (D) ROI from muscle expressing *GalT-YFP* using *MHC-GAL4*; GFP (green) and CLC (magenta). Insets directly below; dotted box muscle ROI, solid box synaptic ROI. (E) ROI as in (D) for muscle expressing *2X-FYVE-GFP* using *BG57-GAL4*. GFP (green) and HRP (magenta). (F) ROI as in (D) for muscle expressing *Pi3K68D-GFP* using *MHC-GAL4*. GFP (green) and CLC (magenta). (G) Quantification of the area of GalT-YFP (Number of synapses: N = 10), FYVE-GFP (Number of synapses: N = 13), or Pi3K68D-GFP puncta (Number of synapses: N = 9), in a WT muscle ROI (black) vs. synaptic ROI (red). (H) Computational slice from EM tomogram of a bouton illustrating invaginations of muscle membrane (black arrows) opposite an active zone (as indicated by the blue arrow showing T-bar). M = mitochondria. Mean ±SEM; ns not significant, *p<0.05, **p<0.01, ***p<0.001, and ****p<0.0001; Student's *t*-test for figure (B and G).

DOI: https://doi.org/10.7554/eLife.31535.014

primary mechanism for generating intracellular pools of PI(3)P that are required for recycling endosome transition to and from the plasma membrane (*Backer, 2008*; *Dall'Armi et al., 2013*; *Juhász et al., 2008*). Importantly, in our unbiased forward genetic screen, we also identified the *Drosophila* orthologue of a class III PI3K (*Vps34*, **Table 1**). This is the only gene in *Drosophila* that encodes a class III PI3K and should, therefore, be required in muscle for recycling endosome formation and function. We asked whether *Drosophila Vps34* is required in muscle for PHP. *Vps34* null mutations are third-instar lethal (*Juhász et al., 2008*). Never-the-less, we confirmed the results of our genetic screen that muscle specific knockdown of *Vps34* blocks PHP (**Figure 7C**). Thus, three independent genes, two of which were derived from the results of a forward genetic screen, highlight an essential role for postsynaptic recycling endosomes in the rapid expression of PHP.

Finally, we thought it important to test whether a generalized disruption of the muscle lipid kinase signaling system might interfere with PHP. Therefore, to complete our analysis, we assessed whether the *Drosophila* class I PI3K might also be necessary, in muscle, for PHP. We found that PHP was normal when we expressed *UAS-RNAi* to knock down class I PI3K (*Pi3K92E*) in muscle (**Figure 7C**). To further address this possibility, we over-expressed the *Pten* phosphatase (*Huang et al., 1999*), which acts to oppose the kinase activity of class I Pi3K. This also had no effect on the induction of PHP (**Figure 7C**). Functionally, the class I PI3K generates $PI(3,4,5)P_3$ and is linked to insulin and TOR signaling (*Knafo and Esteban, 2012*). The lack of an observed effect on the rapid expression of PHP is consistent with prior observations that TOR and S6K act in muscle in a translation-dependent manner to consolidate or maintain expression of PHP, but are dispensable for the rapid expression process (*Penney et al., 2012*).

## A postsynaptic Golgi compartment at the *Drosophila* NMJ

The time-course of PHP induction, occurring in seconds to minutes, implies the existence of mechanisms to control the secretion of retrograde signaling molecules at the postsynaptic side of the active zone. Recent work has demonstrated that Synaptotagmin- 4 and Syntaxin- 4 control the secretion of growth factors at the postsynaptic side of the NMJ (*Akbergenova and Littleton, 2017*; *Harris et al., 2016*; *Rodal et al., 2011*). However, *syt4* mutations were found to have normal PHP (*Dickman and Davis, 2009*). Thus, a separate secretory system must be involved in the rapid expression of PHP.

When examining the distributed Golgi system in *Drosophila* muscle, we discovered that Golgi adopt a distinct morphology within the postsynaptic membranes at the NMJ. In this region, Golgi are statistically significantly smaller and appear more dense than in the surrounding muscle (**Figure 7D,G**). Similarly, puncta of FYVE-GFP and Pi3K68D-GFP are significantly smaller and more densely packed in the subsynaptic region (**Figure 7E–G**). While the functional relevance of this distinction remains unclear, it is clear that the entire secretory system, inclusive of Golgi, and endosomes are concentrated to the postsynaptic membrane system where it is poised to participate in retrograde, homeostatic signaling, among other synaptic functions.

Finally, we have employed 3D TEM tomography to examine the architecture of the SSR to explore the interface of the neuronal and endosomal membrane systems at the NMJ. The postsynaptic muscle membranes at the *Drosophila* NMJ are termed the sub-synaptic reticulum (SSR). The SSR is a complex, multi-layered membrane structure that envelops the nerve terminal. It is well established that endosomal and secretory proteins localize to the SSR and, in some cases, concentrate at this structure (*Akbergenova and Littleton, 2017*). But, because the SSR membrane architecture is so complex (~1 μM thick), it remains unclear how the pre- and postsynaptic membranes of

the nerve terminal interface with the SSR membrane system. Employing 3D TEM tomography (see Materials and methods), we reveal that the postsynaptic plasma membrane often invaginates into the SSR membrane system at sites that are directly adjacent to the active zone. Active zones are defined as sites of electron density between the pre- and postsynaptic plasma membranes, clustered presynaptic vesicles and the presence of a presynaptic T-bar (*Figure 7H*; *Video 1*). The sites of post-synaptic membrane invagination occur directly adjacent to the tightly opposed membranes of the synaptic cleft, creating a postsynaptic membrane that is in direct proximity to the SSR membranes and, by extension, in close proximity to the muscle endosomal and secretory signaling systems. Finally, the synaptic cleft also becomes continuous with the inter-cellular spaces within the SSR. This organization would facilitate the exchange of signaling information between the muscle and synapse, perhaps enabling the type of rapid, homeostatic signaling that is characteristic of PHP.

## Loss of *Pi3K68D* renders the expression of PHP sensitive to changes in extracellular calcium

In order to explore why PHP fails following loss of *Pi3K68D*, we examined the rapid expression of PHP across a range of extracellular calcium concentrations, from 0.3 to 1.5 mM $[Ca^{2+}]_e$. Remarkably, while PHP remained completely blocked at 0.5 mM $[Ca^{2+}]_e$, PHP was restored in the range of 0.7–1.5 mM $[Ca^{2+}]_e$. Indeed, there is a switch-like transition in the expression of PHP between 0.5 and 0.7 mM $[Ca^{2+}]_e$ (*Figure 8A,B*). This is clearly observed by plotting the percent reduction in mEPSP amplitude caused by the application of PhTX versus the percent change in presynaptic release (quantal content), which defines the homeostatic response to PhTX application (*Figure 8B*).

We performed several additional experiments to explore the switch-like calcium sensitivity of PHP in *Pi3K68D* mutants. First, PHP is blocked at 0.3 mM $[Ca^{2+}]_e$ in the *Pi3K68D-GluRIIA* double mutant (see above), but normally expressed at 1.5 mM $[Ca^{2+}]_e$ (*Figure 8C*). Since the *Pi3K68D-GluRIIA* double mutants develop at elevated calcium in vivo (1.5 mM $[Ca^{2+}]_e$ is the assumed average physiological calcium concentration), PHP must have been induced and fully expressed throughout larval life. Thus, loss of postsynaptic *Pi3K68D* must render the presynaptic expression mechanism sensitive to lower concentrations of extracellular calcium. We confirmed this conclusion by incubating the *Pi3K68D* mutant in PhTX at 0.3 mM $[Ca^{2+}]_e$. Then, we immediately switched the preparation to 1.5 mM $[Ca^{2+}]_e$ and found that PHP is fully expressed. Thus, the presynaptic expression of PHP has been rendered acutely calcium sensitive in *Pi3K68D*.

One possible explanation for the switch-like, calcium-dependence of PHP is that loss of *Pi3K68D* is somehow compensated by changes in *Vp34* expression or activity, partially substituting for loss of *Pi3K68D* (note, however, that PI(3)P levels are substantially diminished in *Pi3K68D*, see above). To address this, we generated double heterozygous animals of *Vps34/+* and *Pi3K68D/+*, but find that PHP is normally expressed (data not shown). Next, we removed one copy of *Vps34/+* from the *Pi3K68D* homozygous mutant and, again, we find that PHP is fully expressed at 1.0 mM $[Ca^{2+}]_e$ (*Figure 8—figure supplement 1*). It remains impossible to completely eliminate possible compensation by *Vps34*, since this gene is essential for viability. However, it does not appear that enhanced *Vps34* expression is substituting for the loss of *Pi3K68D*.

Finally, we note that baseline neurotransmitter release appears to be differentially affected by changing extracellular calcium in *Pi3K68D*. Therefore, we explored the possibility that altered release is related to the switch-like effect of external calcium on PHP expression. Presynaptic release is wild-type when recordings are made at 0.5 mM $[Ca^{2+}]_e$ (*Figure 8D*). However, in the range of 0.7–1.5 mM $[Ca^{2+}]_e$ we find a significant deficit in average EPSC amplitude (*Figure 8D*), without a change in mEPSP amplitude (*Supplementary file 1*). Consistent with a

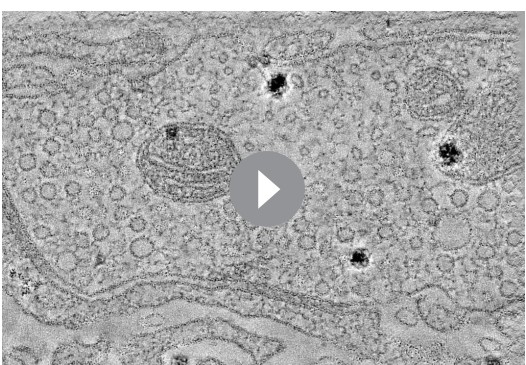

**Video 1.** TEM tomogram of NMJ. An electron microscopy tomogram to examine the NMJ, showing invaginations of muscle membrane opposite an active zone.
DOI: https://doi.org/10.7554/eLife.31535.015

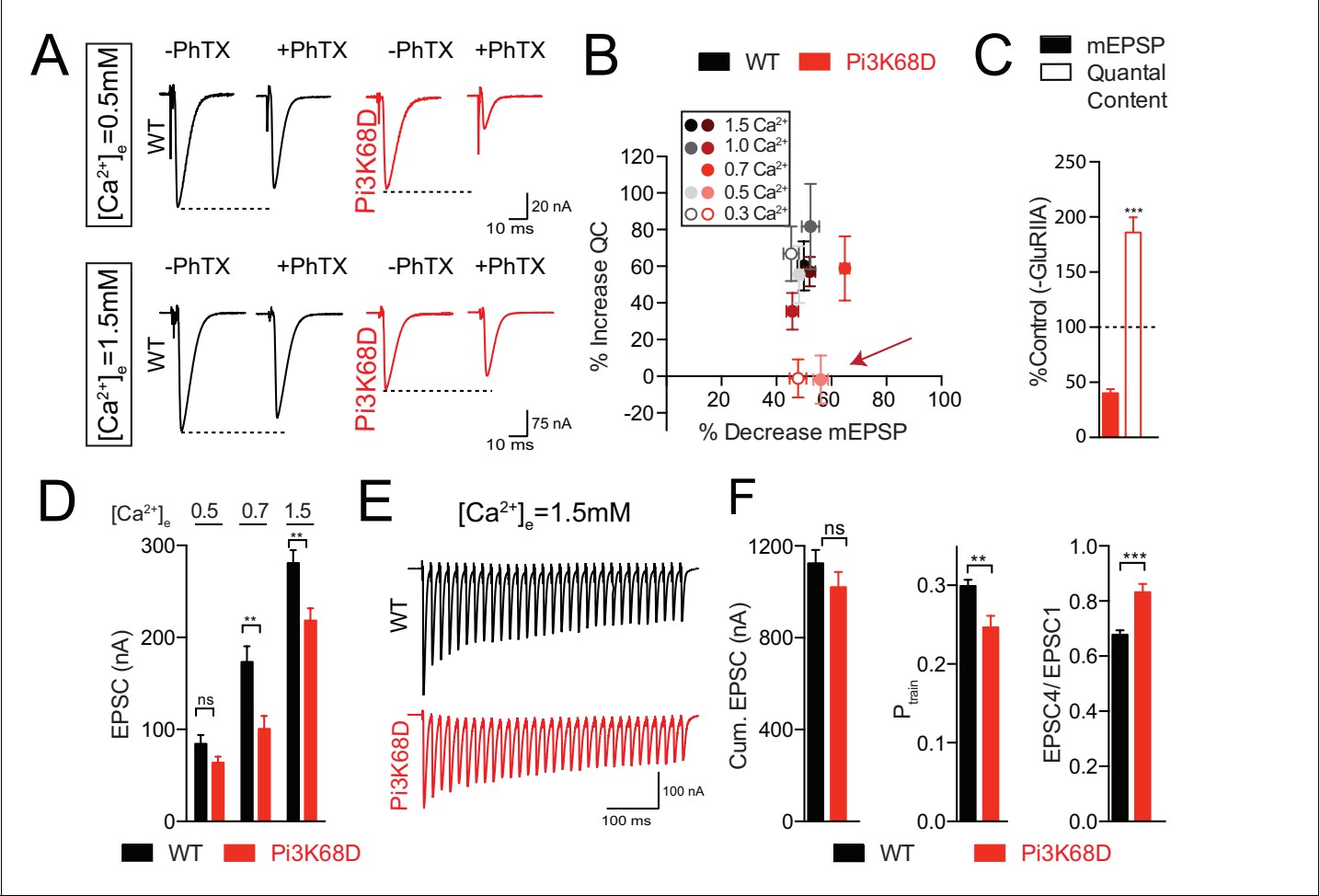

**Figure 8.** Calcium dependence of homeostatic plasticity in *Pi3K68D* mutants. (**A**) Representative traces at 0.5 mM (**A**) and 1.5 mM (**B**) [Ca²⁺]ₑ, in the absence or presence of PhTX as indicated. (**B**) Percent decrease mEPSP due to PhTX application is plotted against the percent increase in quantal content for the indicated calcium concentrations (inset). Red arrow highlights block of PHP at 0.3 and 0.5 mM [Ca²⁺]ₑ. (**C**) Average percent change for mEPSP (filled bars) and quantal content (open bars) for *Pi3K68D AH1;GluRIIAsp16* double mutants compared to *Pi3K68D* mutant alone. Recordings at 1.5 mM [Ca²⁺]ₑ. (**D**) Average EPSC amplitude at 0.5, 0.7, and 1.5 mM [Ca²⁺]ₑ as indicated for WT (black) and *Pi3K68D AH1* (red). (**E**) Representative EPSCs during a stimulus train, calcium as indicated. WT (black) or *Pi3K68D AH1* (red). (**F**) Cumulative EPSC (left), Pₜᵣₐᵢₙ (middle) and EPSC4/EPSC1 (right). WT (black) or *Pi3K68D AH1* (red). Mean ±SEM; ns not significant, *p<0.05, **p<0.01, ***p<0.001 Student's *t*-test.

DOI: https://doi.org/10.7554/eLife.31535.016

The following figure supplement is available for figure 8:

**Figure supplement 1.** Loss of *VPS34* does not enhance the phenotype of *Pi3K68D*.

DOI: https://doi.org/10.7554/eLife.31535.017

change in presynaptic release, we observe a decrease in presynaptic release probability during short, high-frequency stimulus trains (Pₜᵣₐᵢₙ; see Materials and methods) and decreased short-term synaptic depression (*Figure 8E–F*). However, the observed decrease in presynaptic release cannot be causally linked to impaired PHP. Three independent postsynaptic manipulations cause impaired PHP at low extracellular calcium without altering PHP at elevated calcium and without affecting presynaptic release at elevated calcium (*Supplementary file 1*). These manipulations include: (1) postsynaptic expression of the *Pi3K68D* kinase dead transgene, (2) postsynaptic knockdown of Rab11 and (3) postsynaptic knockdown of *Vps34*. In all manipulations, PHP is impaired at 0.3 mM [Ca²⁺]ₑ (*Figures 4E* and *7A*) but restored at 1.5 [Ca²⁺]ₑ with no defect in baseline transmission (*Supplementary file 1*). Note that impaired muscle health caused by muscle over-expression of wild-type *Pi3K68D* precludes two-electrode voltage clamp experiments and, therefore, analysis of post-synaptic rescue of baseline neurotransmission.

## Discussion

We have screened the *Drosophila* kinome and phosphatome for genes that control the rapid expression of PHP. This screen identified three components of a conserved, postsynaptic lipid signaling pathway that is essential for the robust expression of PHP including: (1) class II PI3K, (2) class III PI3K (*Vps34*) and a gene encoding the *Drosophila* orthologue of *PI4K* (not examined in detail in this study). We go on to demonstrate that *Pi3K68D* is essential, postsynaptically for PHP. Pi3K68D resides on a Clathrin-positive membrane compartment that is positioned directly adjacent to Golgi membranes, throughout muscle and concentrated at the postsynaptic side of the synapse. *Pi3K68D* is necessary for the maintenance of postsynaptic PI(3)P levels and the recruitment of Rab11 to intracellular membranes, likely PI(3)P-positive recycling endosomes. Postsynaptic *Rab11* and *Vps34* knockdown block PHP in an unusual, calcium-dependent manner that phenocopies *Pi3K68D*. Thus, we have identified a postsynaptic signaling platform, centered upon the formation of PI(3)P and Rab11-positive recycling endosomes, that is essential for PHP.

We first consider whether postsynaptic Pi3K68D, Vps34 and Rab11 might alter PHP through modulation of postsynaptic glutamate receptor abundance. There is no consistent change in mEPSP amplitude in *Pi3K68D* mutants or following muscle-specific knockdown of Rab11 or Vps34 that could account for altered PHP. Therefore, functionally, there is no evidence for a change in glutamate receptor abundance at the postsynaptic membrane that could drive the phenotypic effects we observe. Anatomically, we also present data examining GluR staining levels. In the *Pi3K68D* mutants, we find no change in GluRIIA levels. GluRIIA subunit containing receptors are the primary mediator of PhTx-dependent PHP (*Frank et al., 2006*). We also report a very modest (16%), though statistically significant, increase in GluRIIB levels. Based on these combined data, it seems unlikely that a change in GluR trafficking is a causal event leading to altered expression of PHP. We note that previous work showed limited GluRIIA receptor mobility within the PSD at the *Drosophila* NMJ (*Rasse et al., 2005*). Thus, we speculate that the function of Pi3K68D, Vps34 and Rab11 during PHP is not directly linked to postsynaptic GluR trafficking.

Any model to explain the role of PI3K, Vps34 and Rab11-dependent endosomal signaling during homeostatic plasticity must account for the phenotypic observation that PHP is only blocked at low extracellular concentrations. More specifically, in animals deficient for *Pi3K68D, Rab11* or *Vps34*, PHP is fully expressed at elevated calcium, following PhTX application or in the *GluRIIA* mutant. However, PHP completely fails when extracellular calcium is acutely decreased (following induction) below 0.7 mM $[Ca^{2+}]_e$. Clearly, the PHP *induction* mechanisms remain fully intact. Instead, the presynaptic *expression* of PHP has been rendered calcium-dependent. It is important to note that PHP can be fully induced in the absence of extracellular calcium, so the concentration of calcium itself is not the defect (*Frank et al., 2009*). In addition, we document trans-heterozygous interactions of *Pi3K68D* with presynaptic *rim* and *dmp*, arguing for the loss of trans-synaptic signaling and a specific function of *Pi3K68D* in the mechanisms of PHP. In very general terms, we conclude a PI3K and Rab11-dependent endosomal signaling platform is necessary to enable the normal expression of PHP. Ultimately, some form of retrograde signaling must be defective due to either: 1) the absence of a retrograde signal that should have normally participated in PHP or 2) the presence of an aberrant or inappropriate signal that dominantly obstructs normal PHP expression. Here, we consider both of these ideas in greater depth.

### Absence of a retrograde signal

First, we consider the possibility that the absence of postsynaptic PI3K and Rab11 signaling could alter the molecular composition or development of the presynaptic terminal due to the persistent absence of a retrograde signal that controls generalized synapse development or growth. Several observations demonstrate that impaired PHP is not a secondary consequence of a general defect in synapse development. We report three independent postsynaptic manipulations (postsynaptic expression of kinase dead *Pi3K68D*, postsynaptic knockdown of *Rab11*, and postsynaptic knockdown of *Vps34*) that have no effect on presynaptic release at any $[Ca^{2+}]_e$, yet block PHP at low $[Ca^{2+}]_e$. In addition, we find no obvious defect in anatomical synapse development (*Figure 3—figure supplement 1*).

Next, we consider the possibility that postsynaptic PI3K and Rab11 signaling eliminate a retrograde signal that is specific for PHP. We recently demonstrated that Semaphorin2b (Sema2b) and

PlexinB (PlexB) define a retrograde signal at the *Drosophila* NMJ that is necessary for PHP (*Orr et al., 2017*). However, both Sema2b and PlexB are essential for the rapid induction of PHP, inclusive of experiments at low and elevated extracellular calcium. Further, acute application of recombinant Sema2b is sufficient to fully induce PHP. Since the induction of PHP remains fully intact in the *Pi3K68D* mutant, and since PHP is rendered calcium sensitive, it suggests that altered Sema2b secretion is not the cause of impaired PHP in the *Pi3K68D* mutant. Never-the-less, this possibility will be directly tested in the future.

## Altered retrograde signaling specificity

Next, we consider the possibility that the loss of PI3K and Rab11 signaling causes aberrant or inap-propriate retrograde signaling, thereby impairing the expression of PHP. This is a plausible scenario because the induction of presynaptic homeostatic plasticity suffers from a common problem inherent to many intra-cellular signaling systems: two incompatible outcomes (1. presynaptic homeostatic potentiation and 2. presynaptic homeostatic depression - PHD) are produced from a common input, and it remains unclear how signaling specificity is achieved. The topic of signaling specificity has been studied in several systems. One system, budding yeast, is a good example. Different phero-mone concentrations can induce several distinct behaviors in budding yeast despite having a com-mon input (pheromone concentration) and underlying signaling systems (*Saito, 2010*; *Schwartz and Madhani, 2004*). Signaling specificity degrades in the background of mutations that affect Map Kinase scaffolding proteins (*Schwartz and Madhani, 2004*). In a similar fashion, presynaptic homeo-static plasticity is induced by a change in mEPSP amplitude. A decrease in mEPSP amplitude causes the induction of PHP, whereas an increase in mEPSP amplitude causes the induction of presynaptic homeostatic depression (PHD) (*Daniels et al., 2004*; *Gaviño et al., 2015*). If a common sensor is employed to detect deviations in average mEPSP amplitude, how is this converted into the specific induction of either PHP or PHD? It has been shown that PHD and PHP can be sequentially induced (*Gaviño et al., 2015*). But, it remains unknown what would happen if the mechanisms of PHP and PHD were simultaneously induced. Under normal conditions this would never occur because mEPSP amplitudes cannot be simultaneously increased and decreased. But, if signaling specificity were degraded in animals lacking postsynaptic PI3K or Rab11, then the expression of PHP and PHD might coincide and create a mechanistic clash within the presynaptic terminal (*Figure 9*).

Signaling and recycling endosomes are, in many respects, ideally suited to achieve signaling spec-ificity during homeostatic plasticity. Signaling specificity can be achieved by mechanisms including sub-cellular compartmentalization of pathways, physically separating signaling elements with protein scaffolds, or through mechanisms of cross-pathway inhibition (*Bardwell et al., 2007*; *Haney et al., 2010*; *Schwartz and Madhani, 2004*). Well-established mechanisms of protein sorting within recy-cling endosomes could physically compartmentalize signaling underlying PHP versus PHD (*Cullen, 2008*; *Grant and Donaldson, 2009*). Alternatively, recycling endosomes can serve as a focal point for signal digitization, integration, and, perhaps, cross-pathway inhibition (*Irannejad et al., 2013*, *2015*; *Villaseñor et al., 2015*; *Villaseñor et al., 2016*). Thus, we propose that the loss of postsynaptic PI3K and Rab11 compromises the function of the postsynaptic endosomal platform that we have identified, thereby degrading homeostatic signaling specificity. As such, this platform could be considered a 'homeostatic controller' that converts homeostatic error signaling into spe-cific, homeostatic, retrograde signaling for either PHP or PHD. One such scenario is proposed in *Figure 9*.

We have also considered other models, but do not favor them. It remains formally possible that the calcium-sensitivity of PHP expression could be explained by a partially functioning PHP signaling system. This seems unlikely given that the same phenotype is observed in four independent genetic manipulations including a null mutation in *Pi3K68D*, postsynaptic expression of kinase dead *Pi3K68D*, postsynaptic knockdown of *Rab11*, and postsynaptic knockdown of *Vps34*. Furthermore, prior experiments examining hypomorphic and trans-heterozygous genetic interactions among essential PHP genes suggest that PHP is either diminished across the entire calcium spectrum or fully functional (*Davis and Müller, 2015*; *Genç et al., 2017*; *Harris et al., 2015*; *Orr et al., 2017*; *Wang et al., 2016*; *Younger et al., 2013*). So, there is no evidence that partial disruption of PHP could account for calcium-sensitive expression of PHP. Finally, our experiments argue against the possibility that compensatory changes in *Vps34* expression partially rescue the *Pi3K68D* mutant phe-notype (*Figure 8—figure supplement 1*).

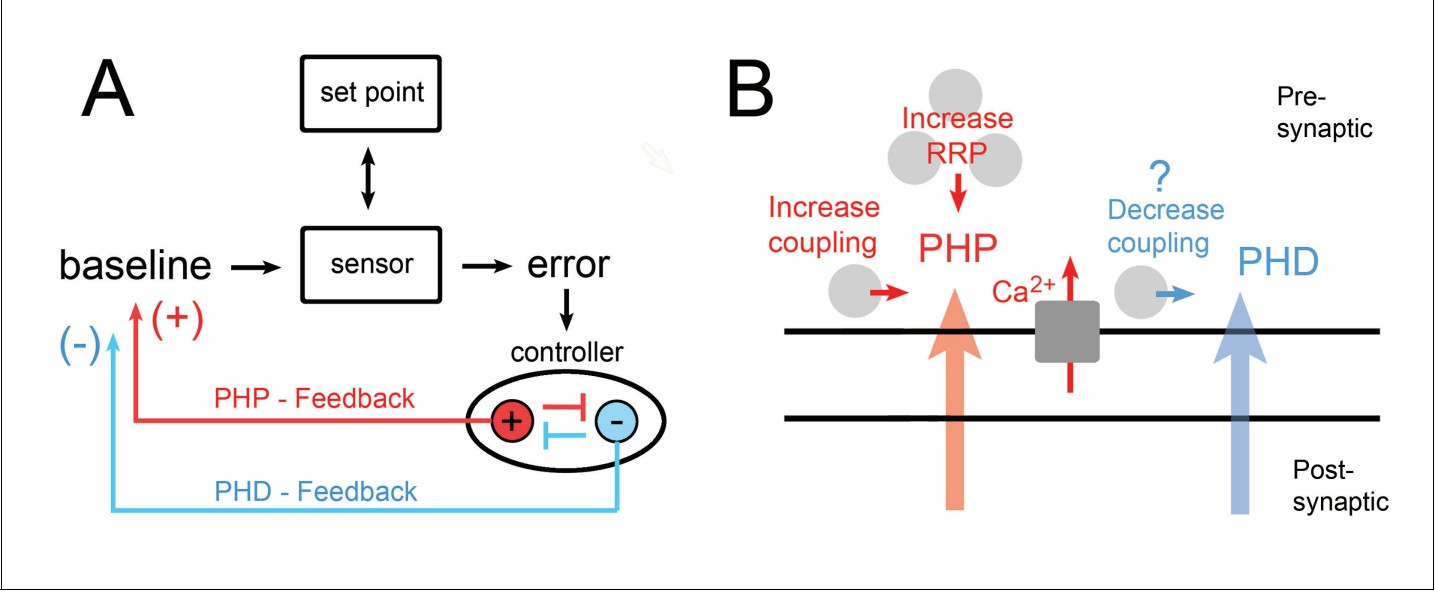

**Figure 9.** Model for the control of PHP and PHD from an endosomal 'controller'. (**A**) A model of homeostatic feedback control inclusive of feedback for both PHP and PHD. The sign for modulation of presynaptic release is indicated. We speculate that signaling pathway cross-inhibition allows for either PHP (red) or PHD (blue) to be selectively induced. When mEPSP amplitudes are decreased (PhTX), this is detected by the sensor and differs from the synaptic set point, causing an error signal to be generated. The error is then relayed to a homeostatic 'controller' where the error is integrated and signaling is induced corresponding to the specific induction of either PHP (red) or PHD (blue). We propose that cross pathway inhibition at the level of the controller allows for the specific induction of either PHP or PHD. We propose that the 'controller' is organized within the PI3K and Rab11-depedent recycling endosomal signaling platform and that loss of this signaling platform leads to inappropriate induction of PHD in the presence of PhTX, causing a mechanistic clash at the level of the presynaptic terminal. (**B**) A model of the neuromuscular junction, highlighting mechanisms of PHP in red (increased calcium influx, increased RRP, and increased vesicle coupling) and PHD in blue, for which very little is understood mechanistically.
DOI: https://doi.org/10.7554/eLife.31535.018

### Postsynaptic Calcium-dependent kinase signaling

We also note another common signaling module that emerged from our genetic screen. Both CamKII and CamKK were identified as potential hits. The identification of CamKII is supported by prior work showing the expression of dominant negative CamKII transgenes disrupt the long-term maintenance of PHP in the *GluRIIA* mutant background (*Haghighi et al., 2003*). It has been assumed that postsynaptic calcium is used to detect the PhTX or GluRIIA-dependent perturbation. But, the logic remains unclear. PHP is induced by diminished GluR function and, therefore, diminished postsynaptic calcium influx (*Newman et al., 2017*). This should diminish activation of CamKII and yet, loss of CamKII blocks PHP. An interesting alternative model is that calcium and calmodulin-dependent kinase activity facilitate the function of the postsynaptic endosomal membrane system. Both calcium and calmodulin are necessary for endosomal membrane fusion (*Colombo et al., 1997*; *Lawe et al., 2003*). In this manner, the action of CamKK and CamKII would be entirely consistent with the identification of Class II/III PI3K and Rab11 as homeostatic plasticity genes.

### Conclusion

We have uncovered novel postsynaptic mechanisms that drive homeostatic plasticity. Eventually, continued progress in this direction may make it possible to not only reveal how stable neural function is achieved throughout life, but to uncover new rules that are essential for the processing of information throughout the nervous system. In particular, PHP has a very large dynamic range, whether one considers data from *Drosophila* or human NMJ or mammalian central synapses. The homeostatic control of presynaptic release can achieve a 7-fold change in synaptic gain, and yet retains the ability to offset even small changes in postsynaptic neurotransmitter receptor function (*Cull-Candy et al., 1980*; *Müller et al., 2015*). Thus, we expect that the regulatory systems that achieve PHP will be complex and have a profound impact on brain function. Here, we have defined

a postsynaptic signaling system responsible for the rapid expression of PHP and propose a novel, albeit speculative, model for the postsynaptic control of PHP, taking into account the need for signaling specificity for the first time. Whether or not we are absolutely correct in proposing how these molecules function within a homeostatic signaling system, their identification paves the way for future advances in understanding how homeostatic signaling is designed and implemented at a cellular and molecular level.

# Materials and methods

**Key resources table**

| Reagent type (species) or resource | Designation | Source or reference | Identifier | Additional information |
|---|---|---|---|---|
| Gene (*Drosophila melanogaster*) | Pi3K68D | NA | FLYB: FBgn0015278 | |
| Gene (*D. melanogaster*) | Rab11 | NA | FLYB: FBgn0015790 | |
| Gene (*D. melanogaster*) | Vps34 | NA | FLYB: FBgn0015277 | |
| Strain/strain background | WT; $w^{1118}$ | NA | $w^{1118}$ | |
| Genetic reagent (*D. melanogaster*) | $GluRIIA^{sp16}$; GluRIIA | (*Petersen et al., 1997*) PMID: 9427247 | FLYB: FBal0085982 | |
| Genetic reagent (*D. melanogaster*) | $elav^{C155}$-GAL4 | BDSC: 458 | FLYB: FBst0000458 | Flybase symbol: P {w[+mW_hs]=GawB} elav[C155] |
| Genetic reagent (*D. melanogaster*) | OK371-GAL4 | (*Mahr and Aberle, 2006*) PMID: 16378756 | | |
| Genetic reagent (*D. melanogaster*) | MHC-GAL4 | (*Petersen et al., 1997*) PMID: 9427247 | | |
| Genetic reagent (*D. melanogaster*) | BG57-GAL4 | (*Budnik et al., 1996*) PMID: 8893021 | | |
| Genetic reagent (*D. melanogaster*) | $rim^{103}$; rim | (*Müller et al., 2012*) PMID: 23175813 | | |
| Genetic reagent (*D. melanogaster*) | $dmp^{f07253}$; dmp | Bloomington Stock Center | BDSC: 19062; FLYB: FBst0019062 | Flybase symbol: w[1118]; PBac{w[+mC]=WH}Mp[f07253] |
| Genetic reagent (*D. melanogaster*) | Pi3K68D-RNAi | Exelixis Collection | HMS:01296 | |
| Genetic reagent (*D. melanogaster*) | UAS-Pi3K68D: GFP | (*Velichkova et al., 2010*) PMID: 20696708 | | |
| Genetic reagent (*D. melanogaster*) | $Vps34^{m22}$ | (*Juhász et al., 2008*) PMID: 18474623 | | |
| Genetic reagent (*D. melanogaster*) | Pi3K68D-MB | Bloomington Drosophila Stock Center | BDSC: 26363; FLYB FBst0026363 http://flybase.org/cgi-bin/uniq.html?FBst0026363%3Efbst | Flybase symbol: w[1118]; Mi{ET1}Pi3K68D[MB08286] CG14131[MB08286] |
| Genetic reagent (*D. melanogaster*) | Pi3K68D-GS | Kyoto Stock Center | KSC: 203158 | Flybase symbol: y[1] w[67c23]; P{w[+mC]=GSV7}GS21729/TM3, Sb[1] Ser[1] |
| Genetic reagent (*D. melanogaster*) | nos-GAL4VP14, UAS-cas9 | (*Port et al., 2014*) PMID: 25002478 | | |
| Genetic reagent (*D. melanogaster*) | PI3K Class I Pi3K92E RNAi | Bloomington Drosophila Stock Center | BDSC: 27690; FLYB: FBst0027690 | Flybase symbol: y[1] v[1]; P{y[+t7.7] v[+t1.8]=TRiP.JF02770} attP2/TM3, Sb[1] |
| Genetic reagent (*D. melanogaster*) | PI3K Class III RNAi; Vps34 RNAi | Bloomington Drosophila Stock Center | BDSC: 33384; FLYB: FBst0033384 | Flybase symbol: y[1] sc[*] v[1]; P{y[+t7.7]v[+t1.8]=TRiP.HMS00261} attP2/TM3, Sb[1] |

*Continued on next page*

*Continued*

| Reagent type (species) or resource | Designation | Source or reference | Identifier | Additional information |
|---|---|---|---|---|
| Genetic reagent (*D. melanogaster*) | *Pten-RNAi* | Bloomington Drosophila Stock Center | BDSC: 33643; FLYB: FBst0033643 | Flybase symbol: y[1] v[1]; P{y[+t7.7] v[+t1.8]=TRiP.HMS00044} attP2TRiP.HMS00044}attP2 |
| Genetic reagent (*D. melanogaster*) | *UAS-GFP-myc-2XFYVE* | Bloomington Drosophila Stock Center | BDSC: 42712; FLYB FBst0042712 | Flybase symbol: w[*]; P{w[+mC]=UAS-GFP-myc-2xFYVE}2 |
| Genetic reagent (*D. melanogaster*) | *UAS-Rab11 RNAi* | Vienna Drosophila RNAi Center | VDRC: 22198; FLYB FBst0454467 | Flybase symbol: w[1118]; P{GD11761}v22198 |
| Genetic reagent (*D. melanogaster*) | *UAS-ManII-GFP* | (*Ye et al., 2007*) PMID: 17719548 | | |
| Genetic reagent (*D. melanogaster*) | *UAS-GalT-YFP* | (*Ye et al., 2007*) PMID: 17719548 | | |
| Genetic reagent (*D. melanogaster*) | *UAS-endostatin* | (*Meyer and Moussian, 2009*) PMID: 19469789 | | |
| Genetic reagent (*D. melanogaster*) | *UAS-endostatin-GFP* | (*Meyer and Moussian, 2009*) PMID: 19469789 | | |
| Genetic reagent (*D. melanogaster*) | *Pi3K68D*[AH1] | This paper | | Indel mutation, premature stop codon at amino acid 1440, made with CRISPR-CAS9 |
| Genetic reagent (*D. melanogaster*) | *UAS-Pi3K68D-Δ21; UAS-Pi3K68D-KD* | This paper | | Generated using site-directed mutagenesis with primers TTTGGAAA CTTTAAGAGAGATC and CATGA TGTTGTCATTGTGG then subsequently cloned into 1100 mCherry. |
| Genetic reagent (*D. melanogaster*) | *UAS-Pi3K68D-ΔN* | This paper | | Primers CACCATGAACGACACC GCCTCCGAC and GTTCCTGGACACC GCGCCC were used to amplify *Pi3K68D-ΔN*, which was then cloned into destination vector 1100 mCherry |
| recombinant DNA reagent | Pi3K68D gRNA | This paper | | ACAGCACTCTGGTACTCGAG for generation of *Pi3K68D*[AH1] |
| recombinant DNA reagent | pCDF3-dU6:3gRNA vector | Addgene | Addgene plasmid #49410 | |
| recombinant DNA reagent | pENTR/D-TOPO | Invitrogen | K240020 | |
| recombinant DNA reagent | destination vector 1100 mCherry | NA | | Gift from Dion Dickman |
| Antibody | anti-BRP (mouse monoclonal) | Developmental Studies Hybridoma Bank | DSHB: nc82 | 1:100, Bouin's fixative |
| Antibody | anti-Discs large; anti-DLG (rabbit) | (*Budnik et al., 1996*) PMID 8893021 | | 1:1,000, Bouin's fixative |
| Antibody | anti-GFP (mouse monoclonal) | Invitrogen | Invitrogen clone 3E6; A-11120 | 1:500, Bouin's fixative |
| Antibody | anti-GluRIIA (mouse monoclonal) | Developmental Studies Hybridoma Bank | DSHB: 8B4D2 (MH2B) | 1:100, Bouin's fixative |
| Antibody | anti-GluRIIB (rabbit polyclonal) | (*Marrus et al., 2004*) PMID 14960613 | | 1:2500, Bouin's fixative |
| Antibody | anti-CLC (rabbit polyclonal) | (*Heerssen et al., 2008*) PMID: 18356056 | | 1:1000, 4% PFA |
| Antibody | Anti-CSP (mouse monoclonal) | (*Zinsmaier et al., 1990*) PMID 2129171 | | 1:250, 4% PFA |
| Antibody | Anti-Syt1 (rabbit polyclonal) | Other | | 1:1000, 4% PFA, gift from Troy Littleton |

*Continued on next page*

*Continued*

| Reagent type (species) or resource | Designation | Source or reference | Identifier | Additional information |
|---|---|---|---|---|
| Antibody | Anti-Rab5 (guinea pig polyclonal) | (*Tanaka and Nakamura, 2008*) PMID: 18272590 | | 1:1000, 4% PFA, gift from Tsubasa Tanaka |
| Antibody | Anti- Rab7 (rabbit polyclonal) | (Tanaka and Nakamura, 2008) PMID: 18272590 | | 1:1000, 4% PFA, gift from Tsubasa Tanaka |
| Antibody | Anti-Rab11 (rabbit polyclonal) | (Tanaka and Nakamura, 2008) PMID: 18272590 | | 1:1000, 4% PFA, gift from Tsubasa Tanaka |
| Antibody | Alexa conjugated secondary antibodies (488, 555, 647) | Jackson Immuno-research laboratories | | 1:500 |
| Sequence based reagent | Primers for sequencing *Pi3K68D* CRISPR mutation | | | GTTTCCAAACATCTGAGCATCG and ATGACTTGCAGCAGGATCAG |
| Software, algorithm | mEPSP analysis | Synaptosoft | Mini Analysis 6.0.0.7 | |
| Software, algorithm | EPSP analysis | (*Ford and Davis, 2014*) | | |
| Software, algorithm | EPSC, Pr, RRP, train analysis | (*Müller et al., 2015*) | | |

## Fly stocks and genetics

All fly stocks were grown at 22–25°C on normal food, except when over-expressing *UAS-Pi3K68D* in the muscle with *BG57-GAL4*, which were grown at 18°C. Fly stocks used are: $w^{1118}$ (wild-type), *GluR-IIA^sp16* (*Petersen et al., 1997*), *elav^C155-GAL4*, *OK371-GAL4* (*Mahr and Aberle, 2006*), *MHC-GAL4* (*Petersen et al., 1997*), *BG57-GAL4* (*Budnik et al., 1996*), *rim^103* (*Müller et al., 2012*), *dmp^f07253* (Bloomington Stock Center 19062), *Pi3K68D-RNAi* (Exelixis Collection HMS01296 - Harvard Medical School), *UAS-Pi3K68D:eGFP* (gift from Amy Kiger), *Vps34^m22* (gift from Tom Neufeld), *Pi3K68D-MB* (Bloomington stock center #26363), *Pi3K68D-GS* (Kyoto Stock Center 203158), *nos-GAL4VP14, UAS-cas9* (*Port et al., 2014*), *PI3K Class I Pi3K92E RNAi* (Bloomington Stock Center #27690), *PI3K Class III RNAi (Vps34)* (Bloomington Stock Center #33384), *Pten-RNAi* (Bloomington Stock Center 33643), *UAS-GFP-myc-2XFYVE* (Bloomington Stock Center 42712), *UAS-Rab11 RNAi* (Vienna Stock Center #22198), *UAS-ManII-GFP* and *UAS-GalT-YFP* (*Ye et al., 2007*), *UAS-endostatin* and *UAS-endostatin-GFP* (*Meyer and Moussian, 2009*).

Using a genomic prep made from transgenic *UAS-Pi3K68D* flies (gift from Amy Kiger), we amplified the *Pi3K68D* cDNA and cloned it into pENTR/D-TOPO (Invitrogen, South San Francisco, CA). This was used as a template to generate *UAS-Pi3K68D-ΔN* and *UAS-Pi3K68D-KDΔ21*. Primers CACCATGAACGACACCGCCTCCGAC and GTTCCTGGACACCGCGCCC were used to amplify *Pi3K68D-ΔN*. The PCR product was cloned into pENTR/D-TOPO (Invitrogen) and then cloned directly into the destination vector 1100 mCherry (gift from Dion Dickman). *UAS-Pi3K68D-KDΔ21* was generated using site-directed mutagenesis with primers TTTGGAAACTTTAAGAGAGATC and CATGATGTTGTCATTGTGG. This was subsequently cloned into 1100 mCherry.

## Generation of CRISPR mutant for *Pi3K68D*

The *Pi3K68D* premature stop mutation was generated following the protocol of (*Kondo and Ueda, 2013*). *Pi3K68D* gRNA was selected using the CRISPR optimal target finder website (http://tools.fly-crispr.molbio.wisc.edu/targetFinder). The gRNA sequence ACAGCACTCTGGTACTCGAG was cloned into the pCDF3-dU6:3gRNA vector (Addgene plasmid #49410, Simon Bullock). Flies expressing the *UAS-gRNA* were crossed with flies expressing *UAS-Cas9* in the germline (*nos-GAL4VP14, UAS-Cas9*). Male offspring were used to create unique stable lines after removing the *UAS-Cas9* and removing in the next generation the *UAS-gRNA*. Putative *Pi3K68D* mutants were sequenced to identify the nature of the Cas9 mediated mutation using the primers GTTTCCAAACATCTGAGCATCG and ATGACTTGCAGCAGGATCAG.

## Electrophysiology

Sharp-electrode recordings and two-electrode voltage clamp recordings were made from muscle six in abdominal segments 2 and 3 from third-instar larvae using an Axoclamp 900A amplifier (Molecular Devices), as described previously (*Frank et al., 2006*; *Müller et al., 2012*). Recordings were made in HL3 saline containing the following components: NaCl (70 mM), KCl (5 mM), $MgCl_2$ (10 mM),

NaHCO$_3$ (10 mM), sucrose (115 mM), trehalose (5 mM), HEPES (5 mM), and CaCl$_2$ (as indicated in figures). For acute pharmacological homeostatic challenge, unstretched larva were incubated in Phi-lanthotoxin-433 (PhTX; 15 µM; Sigma-Aldrich) for 10 min. (*Frank et al., 2006*). Recordings were excluded if the resting membrane potential (RMP) was more depolarized than −60 mV, except for over-expression of *UAS-PTEN* and *UAS-Pi3K68D* which uniformly compromised RMP. A threshold 40% decrease in mEPSP amplitude, below average baseline, was used to confirm the activity of PhTX. For *UAS-Rab11-RNAi* expression a 15% decrease was used. EPSP traces were analyzed in IGOR Pro (Wave-Metrics) and with previously published routines in MATLAB (Mathworks) (*Ford and Davis, 2014*). mEPSP traces were analyzed using MiniAnalysis 6.0.0.7 (Synaptosoft), averaging at least 100 individual mEPSP events. EPSC amplitudes were analyzed in IGOR Pro (Wave-Metrics) with previously routines (*Müller et al., 2015*). Quantal content was calculated by dividing mean EPSP by mean mEPSP. The RRP was estimated by cumulative EPSC analysis, as described previously (*Müller et al., 2012*; *Schneggenburger et al., 1999*). In brief, muscles were clamped at −65 mV in two-electrode voltage clamp during a stimulus train (60 Hz, 30 stimuli). RRP for each muscle was calculated by dividing cumulative EPSC amplitude in TEVC by mEPSP amplitude in current clamp. P$_{train}$ was calculated by dividing mean first EPSC amplitude by mean cumulative EPSC. Best-fit curves for mEPSP amplitude versus quantal content were fit in Prism 6 (GraphPad) using a power function for all wild-type data points ± PhTX. 95% data intervals were fit in IGOR Pro (Wave-Metrics) using a power function.

## Immunohistochemsitry

Standard immunohistochemistry was performed as previously described (*Pielage et al., 2005*). In brief, filleted third instar larvae were fixed in either Bouin's fixative (Sigma-Aldrich, 5 min) or 4% PFA (Affymetrix, 30 min), as indicated for each antibody below. Preps were washed in PBT (PBS with 0.1% Triton) for 1 hr, then incubated overnight at 4° in primary antibody in PBT. Larval fillets stained for the following primary antibodies were fixed with Bouin's: mouse anti-BRP (1:100, Developmental Studies Hybridoma Bank, [*Kittel et al., 2006*]), rabbit anti-Discs large (Dlg, 1:1,000, [*Budnik et al., 1996*]), mouse anti-GFP (1:500, Invitrogen clone 3E6), mouse anti-GluRIIA (Developmental Studies Hybridoma Bank [*Marrus et al., 2004*]), rabbit anti-GluRIIB (1:2500, a gift from Aaron DiAntonio [*Marrus et al., 2004*]). Larval fillets stained for the following primary antibodies were fixed with PFA: rabbit anti-CLC (1:1000 [*Heerssen et al., 2008*]), mouse anti-CSP (1:250 [*Zinsmaier et al., 1990*]), rabbit anti-Syt1 1:1000 (a gift from Troy Littleton), guinea pig anti-Rab5 1:1000 (a gift from Tsubasa Tanaka), rabbit anti-Rab7 1:1000 (a gift from Tsubasa Tanaka), rabbit anti-Rab11 1:1000 (a gift from Tsubasa Tanaka). Preps were washed in PBT for 1 hr and incubated in secondary antibody in PBT for 1 hr at room temperature. Alexa-conjugated secondary antibodies were used at 1:500 and FITC-, Cy3- and Cy5-conjugated HRP was used at 1:100 (Jackson Immuno-research Laboratories). Preps were mounted in Vectashield (Vector). Immunolabeling of surface GFP was performed as described in (*Wang et al., 2014*) by incubating larval preparations in rabbit anti-GFP antibody (1:500) before permeabilization of cell membranes.

## Image acquisition and analysis

Deconvolution wide field imaging for synapse morphology was performed using a 100x (1.4 NA) plan Apochromat objective (Carl Zeiss) on an Axiovert 200 inverted microscope (Carl Zeiss) equipped with a cooled CCD camera (CoolSNAP HQ; Roper Scientific). Image acquisition and analyses were performed in SlideBook software (Intelligent Imaging Innovation). Structured illumination fluorescence microscopy was performed using an N-SIM System (Nikon) with an Apo TIRF 100x/1.49 oil objective on a Ti-E microscope (Nikon) and an Andor DU897 camera. Z-stacks of 120 nm were collected for muscle 4 or 6/7. Images were reconstructed in NIS-Elements 4.12. Maximum projection images were made.

## NMJ morphology analysis

Quantification of BRP and bouton number was performed as previously described (*Wang et al., 2014*). Boutons were counted manually on a Zeiss axioskop 40 compound microscope (40x, 1.1nA lens). Boutons of type 1b and 1 s were independently quantified for abdominal segments A2 and A3. Active zone number was calculated by counting individual BRP puncta (100x, 1.4nA lens) from

maximum intensity projection deconvolved images, as previously described (*Wang et al., 2014*). Synaptic Clathrin Light Chain was quantified by masking for the neuronal membrane with HRP, then quantifying average fluorescence intensity using Slidebook Software in maximum projection images (Intelligent Imaging Innovation). Syt1 and CSP were quantified as previously described (*Harris et al., 2015*).

FYVE-GFP, CLC, Rab7, Rab11, GalT-YFP, Pi3K68D-GFP, and FYVE-GFP puncta number and size were quantified with maximum projections from a fixed number of image planes. A region of interest in muscles 6, segments 2 and 3, was chosen not inclusive of muscle nuclei. Images were thresholded to the same value and puncta were analyzed with Fiji (*Schindelin et al., 2012*). The number of puncta was counted per ROI, and each puncta area was also measured. Rab5, Rab7, and Rab11 mean intensity per ROI were quantified by taking sum projections of the same number of slices of images (Fiji).

## Electron microscopy tomography

Third instar $w^{3605}$ larvae were prepared for electron microscopy as described in (*Harris et al., 2015*). For EM tomography, 200 nm sections cut with a Diatome diamond knife using a Leica UC-T ultramicrotome were picked up on Pioloform films with 2 nm C on Synaptek slot grids (Ted Pella, Inc). Sections were post-stained with 7.5% uranyl acetate followed by Sato's lead sa (*Sato, 1968*). Dual-axis tilt series images (±60 deg) were acquired with an FEI T20 electron microscope at 200 kV equipped with a Tietz F816 digital camera. Tomograms were reconstructed using the eTomo package in IMOD (*Mastronarde, 1997*).

## Acknowledgements

We thank Özgür Genç and Brian Orr for critical reading of the manuscript, and Amy Kiger and Flora Rutaganira (Kevan Shokat Lab) for collegial support throughout. Supported by NIH Grant R35NS097212 to GWD and 5T32GM007618 and F30NS092211 to AGH. We acknowledge support of the Nikon Imaging Center at UCSF and the Kavli Institute for Fundamental Neuroscience at UCSF.

## Additional information

### Competing interests

Graeme W Davis: Reviewing editor, *eLife*. The other authors declare that no competing interests exist.

### Funding

| Funder | Grant reference number | Author |
| --- | --- | --- |
| National Institute of Neurological Disorders and Stroke | R35NS097212 | Graeme W Davis |
| National Institute of General Medical Sciences | 5T32GM007618 | Anna G Hauswirth |
| National Institute of Neurological Disorders and Stroke | F30NS092211 | Anna G Hauswirth |

The funders had no role in study design, data collection and interpretation, or the decision to submit the work for publication.

### Author contributions

Anna G Hauswirth, Conceptualization, Data curation, Formal analysis, Funding acquisition, Investigation, Visualization, Methodology, Writing—original draft, Writing—review and editing; Kevin J Ford, Conceptualization, Data curation, Software, Formal analysis, Validation, Investigation; Tingting Wang, Data curation, Formal analysis, Investigation, Writing—review and editing; Richard D Fetter, Conceptualization, Data curation, Formal analysis, Validation, Investigation, Visualization, Writing—review and editing; Amy Tong, Data curation, Formal analysis, Validation, Methodology; Graeme W

Davis, Conceptualization, Supervision, Funding acquisition, Methodology, Writing—original draft, Project administration, Writing—review and editing

**Author ORCIDs**
Graeme W Davis http://orcid.org/0000-0003-1355-8401

**Decision letter and Author response**
Decision letter https://doi.org/10.7554/eLife.31535.022
Author response https://doi.org/10.7554/eLife.31535.023

## Additional files

### Supplementary files

• Supplementary file 1. Source data for all electrophysiological data for all figures and supplemental figures. Data includes mEPSP amplitudes, EPSP or EPSC amplitudes, quantal contents, and sample size (N).
DOI: https://doi.org/10.7554/eLife.31535.019

• Transparent reporting form
DOI: https://doi.org/10.7554/eLife.31535.020

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
