## [Decision Letter]

Thank you for submitting your article "A postsynaptic Pi3K-CII dependent signaling controller for presynaptic homeostatic plasticity" for consideration by *eLife*. Your article has been reviewed by three peer reviewers, and the evaluation has been overseen by Kang Shen as the Reviewing Editor and Eve Marder as the Senior Editor. The following individual involved in review of your submission has agreed to reveal his identity: Joshua M Kaplan (Reviewer #1).

The reviewers have discussed the reviews with one another and the Reviewing Editor has drafted this decision to help you prepare a revised submission.

All three reviewers recognized that this paper is the first comprehensive characterization of the postsynaptic mechanisms of PHP and found it appropriate for publication in *eLife*. They did raise a number of questions which I will list below. Through discussion with each other, we have come to the conclusion that you can address the issues by revising the text and including new discussions. If you have additional data that speak to some of the questions such as mutant allele analysis of vps34 or rab11, you are also welcome to add them. In particular, the reviewers would like to see some discussions about the steady state GluR levels in the endocytic mutants and how that could potentially affect PHP.

We expect a quick turn around after you resubmit your manuscript since the reviewers did not demand specific experiments to be added. We look forward to receiving a revised version of this manuscript.

Summary:

The manuscript by Hauswrith et al. describes the results of an RNAi screen of the *Drosophila* kinase and phosphatase proteome to identify components of homeostatic plasticity at the fly NMJ. The authors identified several hits in the late endosomal pathway, and focus on the characterization of one of those hits in detail – PI3K-c11. The authors provide compelling evidence that a postsynaptic late endosomal pathway is essential for homeostatic plasticity, and show that PI3K-c11 mutants disrupt the formation of this compartment. They find an interesting effect on extracellular calcium levels as a regulator of the mechanism, indicating that the pathway is required to support homeostatic at low calcium levels, but not high. They provide some interesting speculation on how this might arise, but sorting it out will require more experimentation. Overall, it's a nice study with some compelling data. I especially liked the individual recordings shown in Figure 1 and Figure 2 that really highlight how robust this form of plasticity is. The characterization of the PI3K-c11 mutant is robust and provides a strong argument for this pathway regulating late endosomal compartments and the homeostatic response to both acute and long-term blockage of postsynaptic glutamate receptors.

Essential revisions:

Please discuss if the steady state GluR levels is affected the endocytic mutants and how that could potentially affect PHP.

Minor points:

1) In Figure 2, the QC scatter plot for Pi3K68D mutants seem substantially shifted to lower values than the predicted curve for WT controls. Does this suggest that QC is reduced at baseline in these mutants?

2) The conclusions concerning Rab11 and VPS34 would be strengthened if analysis of mutant alleles (or mosaic animals) were provided. I agree that the strength of the Pi3K86D data offset this concern somewhat; however, analysis of mutant alleles would strengthen the authors' conclusions. Perhaps you already have such data?

3) Could the authors speculate about the identity of the EE/RE cargo that requires Pi3K68D?

4) Do the endosomal mutations and RNAi alter RE/EE cargo flux? This could be measured with a fluorescent tracer (e.g. Transferrin).

5) Wouldn't you expect changes in GluR abundance if RE trafficking is perturbed? The data shown in Figure 4 suggest that mEPSP amplitudes are decreased with Pi3K68D over-expression. Was the GluR staining also decreased in these animals?

6) What is the Muscle O/E genotype shown in Figure 4? I thought that this UAS was lethal when expressed by the BG57 GAL4 driver.

7) The Discussion has an elaborate section involving speculation about PHD. Does Pi3K86D regulate PHD? If not, this aspect of the discussion seems weakly connected to the main findings reported here and should be minimized.

8) Is it fair to normalize EPSCs to mEPSPs for QC calculations?

9) The authors should discuss/speculate about why PHP in Pi3K mutants is sensitive to external calcium levels? Is PHP sensitive to EGTA-AM?

10) Does Pi3K68D mutation alter RMP? (over-expressing Tgene does, subsection “Pi3K68D is required postsynaptically for PHP”, first paragraph).

11) Is FYVE-GFP or Rab11 staining altered when PHP is induced? This would suggest that post-synaptic Endosomes signaling are involved in PHP induction.

12) At several points in the text, you refer to a post-synaptic endosomal "platform" for PHP (e.g. when discussing the model in Figure 9). Please explain what you mean by this term.

13) Pi3K68D-MB is apparently the Minos insertion MB08286 (Materials and methods). I am surprised this has a mutant phenotype, as the splice trap in the Minos is in the wrong orientation for Pi3K68D. This mutation would be more likely to block expression of CG14131, a gene within the Pi3K68D intron that goes in the opposite direction. The MB is in its intron. They should look at the MiMIC insertion nearby (MI15179, also available from Bloomington), as that is in the right orientation to trap splicing of Pi3K68D. Of course, their CRISPR allele appears to show that the phenotype can be produced by only perturbing expression of Pi3I68D, so this should not call any of their results into question. However, the MiMIC allele would be useful for another experiment they should do (although not necessarily for this paper), which I will describe below.

14) I question their interpretation of the deltaN overexpression phenotype (subsection “Pi3K68D is required postsynaptically for PHP”, last paragraph). If the N terminus interacts with clathrin, and they overexpress a mutant lacking this region in a wt background, this protein (if it is active) should still have activity but be incorrectly localized. In their model, it should not interfere with binding of the wt protein (which is still expressed) to clathrin, since deltaN lacks the clathrin binding site. If clathrin does indeed bind to the N terminus, and they wish to test whether this is important, they would need to instead ectopically express only the N-terminal region, without the kinase. This should then act as a dominant negative, occupying all the sites on clathrin and thus preventing correct localization of the wt protein. Actually, the fact that overexpression of the kinase-dead mutant blocks PHP already makes this point (second paragraph of aforementioned subsection), although not as cleanly as expression of only the N terminus would. The KD mutant should act as a DN, unlike the deltaN mutant, because it would bind to clathrin, occupying the binding sites for the wt kinase, but would not itself have kinase activity.

15) In this regard, it is interesting that there are two forms of the Pi3K68D mRNA (RE and RB) that would encode truncated proteins (PE and PB) encoding only the N terminus. Perhaps these are natural regulators of the kinase whose expression determines whether or not the kinase can find a clathrin binding site? It would be interesting to see where these isoforms are expressed, and to express RE or RB cDNAs and see if they act as DNs.

16) I don't understand the section on the effects of changing Ca concentration. The results are intriguing but seem difficult to interpret in any mechanistic way given the information that exists at present. The Discussion is very hard to read. Perhaps this is inevitable given the confusing nature of the results.

17) Related to point 16), they seem to be saying that since PHP works normally in the mutant at physiological (1.5 uM) Ca++ concentrations (subsection “Loss of Pi3K68D renders the expression of PHP sensitive to changes in extracellular calcium”, second paragraph), these mutants must have been normal with respect to this phenomenon during their development in vivo. It is only when they are filleted and incubated at low Ca++ that a phenotype is observed. It would be very informative about what is going on if they could determine whether the development of the larva in the absence of Pi3K68D renders fillets from these animals abnormal with regard to PHP at low Ca++, or if the activity of Pi3K68D during the experiment is required for normal PHP at low Ca++. It seems like this should be addressable in their system, because there are many Pi3K inhibitors that have been described. They could fillet wt embryos, or embryos lacking one copy of the gene, and then incubate them with the inhibitors and see if this affects PHP. But, perhaps the available inhibitors are not specific enough for this form of Pi3K, or perhaps they don't work in *Drosophila*? It seems like the investigators probably know the answer to this question already, since it is an obvious experiment. If so, they should inform us in the paper. If not, they should do the experiment.

18) If inhibitors don't work or are not specific enough, there is another approach that could address this question. This experiment would require more time than is normally given for revisions, and the paper is already strong, so I would not require it for publication. The Bellen lab (Nagarkar-Jaiswal et al. *eLife* 2015) has shown that the DeGradGFP method can transiently inhibit gene function by degrading a GFP fusion of the protein. There is a MiMIC in an appropriate position to allow replacement of the MiMIC by GFP using their methods, to create a "protein trap". In the Bellen paper, they show that such protein traps often retain function. Thus, if they were to make a protein trap and found that this allele behaved like the wt protein, they could then turn off Pi3K68D activity using the temperature-sensitive DeGradGFP mechanism in larvae that had developed with normal activity. In the Bellen paper, they show that they can rapidly shut off Dunce and turn it back on by temperature shifts.

19) From the images, it’s clear there are robust changes in postsynaptic endosomal compartments. However, it is less clear if there is any change going on presynaptically. Clearly they demonstrate the effect on homeostatic plasticity is postsynaptic, but it would be interesting to know if the same changes in subcellular markers is occurring in the presynaptic bouton. I assume PI(3)P would be required in both compartments to properly generate that endosomal compartment – if not it would be interesting, but very surprising. I assume the authors already have that data from their images, so would be nice to comment on it in the text.

20) The authors nicely show postsynaptic rescue of the Pi3K mutant phenotype. However, it wasn't clear why they didn't also show postsynaptic RNAi knockdown blocks PHP. Did I miss that somewhere – if not, why not show that as well to complement the rescue?

21) The last minor comment deals with the Pi3K68D structure function part. They infer the clathrin binding domain and kinase activity are important by overexpressing constructs missing these elements and observing a block in PHP. I would have preferred to see this being done as a rescue of the null mutant – seems a far easier result to interpret. Why was the overexpression used instead of rescue?

22) Not sure how the EM was quantified, or if it was.

---

## [Author Response]

Essential revisions:Please discuss if the steady state GluR levels is affected the endocytic mutants and how that could potentially affect PHP.

In our original submission, we did provide experimental evidence arguing against a change in GluR levels being causal. These data were presented in the section of the Results subtitled, “*Pi3K68D* mutants have normal morphology and glutamate receptor abundance”. However, we did not previously comment on these data other than to conclude the following: “In conclusion, we find no evidence for a substantial decrease in key presynaptic proteins, postsynaptic neurotransmitter receptors, or bouton numbers in the *Pi3K68D* mutant.”

In response to this reviewer request, we have now added the following paragraph to the beginning of our Discussion section, inclusive of a new reference documenting limited mobility of the GluR receptors once incorporated into the PSD at the *Drosophila* NMJ. Newly added text to our Discussion is as follows: *“*We first consider whether postsynaptic Pi3K68D, Vps34 and Rab11 might alter PHP through modulation of postsynaptic glutamate receptor abundance. […] Thus, we speculate that the function of Pi3K68D, Vps34 and Rab11 during PHP is not directly linked to postsynaptic GluR trafficking.”

Minor points:1) In Figure 2, the QC scatter plot for Pi3K68D mutants seem substantially shifted to lower values than the predicted curve for WT controls. Does this suggest that QC is reduced at baseline in these mutants?

There is, indeed, a small reduction in average QC recorded at 0.3mM calcium. WT QC=30.3 ± 1.2 while *Pi3K68D* is 26.1 ± 1.2, with p=0.018. As shown in Figure 7 of our original submission, this defect in baseline QC is also observed when measuring synaptic currents at 0.5 and 1.5mM extracellular calcium. Thus, all of our data are internally consistent and consistent with our original conclusions. We have modified the text to point out this small change in baseline average quantal content in Figure 2. We thank the reviewer for prompting us to make this addition to the text. The text now reads, “We also note that there is a small but significant decrease in baseline QC (minus PhTX) in the *Pi3K68D* mutant background (WT QC=30.3 ± 1.2 and *Pi3K68D* QC=26.1 ± 1.2; p=0.018; see below for further analysis and discussion of baseline transmission).”

2) The conclusions concerning Rab11 and VPS34 would be strengthened if analysis of mutant alleles (or mosaic animals) were provided. I agree that the strength of the Pi3K86D data offset this concern somewhat; however, analysis of mutant alleles would strengthen the authors' conclusions. Perhaps you already have such data?

In principle, we agree. But, there are several reasons why we have restricted our analyses to UAS-RNAi. First, *Vps34* null mutations are not viable to the 3^rd^ instar stage and mosaic analysis in poly-nucleated muscle is problematic. Second, we needed to specifically test the function of Rab11 and Vps34 in muscle, without altering important neuronal function of these molecules. Tissue-specific RNAi is the method of choice. Finally, in our original submission, we did go so far as to remove one copy of *Vps34* in the *Pi3K68D* homozygous background and found no additive effect (Figure 8—figure supplement 1).

3) Could the authors speculate about the identity of the EE/RE cargo that requires Pi3K68D?

Yes. We have added new text. First, we address the possibility that the EE/RE cargo could involve Semaphorin2b, which we recently identified as a retrograde signaling molecule (Orr et al., 2017). We now state in the Discussion section, “Next, we consider the possibility that postsynaptic PI3K and Rab11 signaling eliminate a retrograde signal that is specific for PHP. […] Never-the-less, this possibility will be directly tested in the future.”

Second, we now provide new data that the EE/RE cargo is unlikely to be Multiplexin, a matrix protein, required for PHP (Wang et al., 2015), that can be cleaved to release Endostatin. In new supplemental information we demonstrate that muscle-specific expression of *UAS-Endostatin* is secreted from the muscle in the *Pi3K68D* mutant, and this condition fails to rescue homeostasis in the *Pi3K68D* mutant (Figure 5—figure supplement 1). These data argue that the block of PHP in the *Pi3K68D* mutant cannot due to a failure to secrete Endostatin from muscle.

4) Do the endosomal mutations and RNAi alter RE/EE cargo flux? This could be measured with a fluorescent tracer (e.g. Transferrin).

We agree this is an interesting question and something to test in the future. It seems likely that the flux of a fluorescent tracer would be altered, but creating a link to the mechanisms of PHP will be a challenge best addressed in a new study.

5) Wouldn't you expect changes in GluR abundance if RE trafficking is perturbed? The data shown in Figure 4 suggest that mEPSP amplitudes are decreased with Pi3K68D over-expression. Was the GluR staining also decreased in these animals?

We did not see changes in GluR abundance (see above) or in mEPSP amplitude in the *Pi3K68D* mutants. The decrease in mEPSP amplitude following Pi3K68D over-expression is most likely caused by impaired muscle resting membrane potential. We refer to the following text in our original submission:

“Muscle recordings from these rescue animals revealed impaired muscle resting membrane potentials (RMP) due to *Pi3K68D* overexpression (RMP = -60.1 ± 1.3 mV without PhTX and -59.3 ± 2.4 mV with PhTX compared to the *Pi3K68D* mutant: -68.2 ± 1.3 mV without PhTX and -68.0 ± 1.8 mV with PhTX). We find an associated decrease in mEPSP amplitude, likely due to diminished driving force (Figure 4).”

6) What is the Muscle O/E genotype shown in Figure 4? I thought that this UAS was lethal when expressed by the BG57 GAL4 driver.

The reviewer is correct. Muscle expression using the *BG57-GAL4* driver is lethal, as we originally stated. However, for this experiment, we used the *MHC-GAL4* reagent to drive *UAS-Pi3K68D-GFP* (see Supplementary file 1 for full genotypes). This animal is viable. The *MHC-GAL4* driver initiates expression post-embryonically, whereas *BG57-GAL4* initiates expression during embryonic development of the musculature. This may account for why *MHC-GAL4* can be used for this experiment. We have amended the text to highlight the use of *MHC-GAL4* and thank this reviewer for pointing out this omission.

7) The Discussion has an elaborate section involving speculation about PHD. Does Pi3K86D regulate PHD? If not, this aspect of the discussion seems weakly connected to the main findings reported here and should be minimized.

We have taken the comments of the reviewers to heart and shortened this section of the Discussion. The ideas in this part of the Discussion are pure speculation. But, as a model, we felt it was valuable to incorporate since it might spur other laboratories to consider these possibilities. We were (and still are) careful to state that this is a speculative model.

8) Is it fair to normalize EPSCs to mEPSPs for QC calculations?

This is simply a means to illustrate the effect of diminished mEPSP amplitude caused by application of PhTX, a technique that we have used in the past. The raw values underlying all of our measurements are presented in the supplementary file so that it is clear how the calculations are made and what the underlying measurements reveal.

9) The authors should discuss/speculate about why PHP in Pi3K mutants is sensitive to external calcium levels? Is PHP sensitive to EGTA-AM?

We devoted a significant portion of our original Discussion section to this topic. Indeed, the second paragraph of the Discussion begins, “Any model to explain the role of PI3K-cII and Rab11-dependent endosomal signaling during homeostatic plasticity must account for the phenotypic observation that PHP is only blocked at low extracellular calcium concentrations.” Following was a rather lengthy discussion of the data and results and speculation. As requested, we have condensed the speculative portion of the text.

10) Does Pi3K68D mutation alter RMP? (over-expressing Tgene does, subsection “Pi3K68D is required postsynaptically for PHP”, first paragraph).

The *Pi3K68D* mutation does not impair RMP. For the data plotted in Figure 2, there is a small but significant increase in RMP in *Pi3K68D*: WT -64.5 ± 0.45 mV and *Pi3K68D* -66.4 ± 0.55 mV, p=0.015.

11) Is FYVE-GFP or Rab11 staining altered when PHP is induced? This would suggest that post-synaptic Endosomes signaling are involved in PHP induction.

It seems unlikely that flux through the endosomal system could be resolved by changes in the steady state levels of these proteins. With this said, future studies to identify potential cargo and flux of this cargo through the endosomal system will be a focus.

12) At several points in the text, you refer to a post-synaptic endosomal "platform" for PHP (e.g. when discussing the model in Figure 9). Please explain what you mean by this term.

A platform can be defined as a surface that is used to support a structure or device. It can also be defined as a vehicle that is used to deliver a cargo or device (definitions are paraphrased from the Merriam Webster dictionary). This term seemed to be a reasonable, generalized way to refer to the function of the postsynaptic endosomal signaling system during homeostatic plasticity.

13) Pi3K68D-MB is apparently the Minos insertion MB08286 (Materials and methods). I am surprised this has a mutant phenotype, as the splice trap in the Minos is in the wrong orientation for Pi3K68D. This mutation would be more likely to block expression of CG14131, a gene within the Pi3K68D intron that goes in the opposite direction. The MB is in its intron. They should look at the MiMIC insertion nearby (MI15179, also available from Bloomington), as that is in the right orientation to trap splicing of Pi3K68D. Of course, their CRISPR allele appears to show that the phenotype can be produced by only perturbing expression of Pi3I68D, so this should not call any of their results into question. However, the MiMIC allele would be useful for another experiment they should do (although not necessarily for this paper), which I will describe below.

In general, the presence of a large transposon insertion within a gene locus can function as a gene mutation. There are many reasons why this is the case, irrespective of whether the transposon functions as a gene-trap. We refer the reviewer to an earlier gene trap paper that we published in which we manipulate the size of the transposon insertion and demonstrate that the size of the transposon affects mutagenicity (Clyne et al., 2003; Genetics). As the reviewer points out, this is just one part of our genetic argument, which includes a newly generated CRISPR mutation, a second transposon insertion allele, tissue-specific RNAi and use of both a kinase-dead, dominant negative transgene as well as an N-terminal truncation.

14) I question their interpretation of the deltaN overexpression phenotype (subsection “Pi3K68D is required postsynaptically for PHP”, last paragraph). If the N terminus interacts with clathrin, and they overexpress a mutant lacking this region in a wt background, this protein (if it is active) should still have activity but be incorrectly localized. In their model, it should not interfere with binding of the wt protein (which is still expressed) to clathrin, since deltaN lacks the clathrin binding site. If clathrin does indeed bind to the N terminus, and they wish to test whether this is important, they would need to instead ectopically express only the N-terminal region, without the kinase. This should then act as a dominant negative, occupying all the sites on clathrin and thus preventing correct localization of the wt protein. Actually, the fact that overexpression of the kinase-dead mutant blocks PHP already makes this point (second paragraph of aforementioned subsection), although not as cleanly as expression of only the N terminus would. The KD mutant should act as a DN, unlike the deltaN mutant, because it would bind to clathrin, occupying the binding sites for the wt kinase, but would not itself have kinase activity.

These are reasonable points, and they highlight that we do not yet know precisely how the delta-N transgene is working. We have primarily used this reagent to underscore the postsynaptic requirement of the Pi3K68D gene. As the reviewer points out, the expression of the kinase dead transgene also addresses this issue. This is a second reagent, and supportive of our argument. It is worth pointing out that the reviewer has presented one scenario, but not the only one. For example, there is evidence that Clathrin binding to the N-terminus facilitates activation of the kinase (Gaidarov et al., 2001). If so, this transgene could function as a dominant interfering reagent, just as expression of the kinase dead transgene does. We were quite careful regarding our interpretation of this experiment in the original Results section of our paper. We stated, “We conclude that either proper kinase localization or regulation, potentially via Clathrin binding, is essential for robust homeostatic signaling. Collectively, these results demonstrate a required postsynaptic function of *Pi3K68D* for presynaptic homeostatic plasticity.”

15) In this regard, it is interesting that there are two forms of the Pi3K68D mRNA (RE and RB) that would encode truncated proteins (PE and PB) encoding only the N terminus. Perhaps these are natural regulators of the kinase whose expression determines whether or not the kinase can find a clathrin binding site? It would be interesting to see where these isoforms are expressed, and to express RE or RB cDNAs and see if they act as DNs.

This would, indeed, be worth examining in the future.

16) I don't understand the section on the effects of changing Ca concentration. The results are intriguing but seem difficult to interpret in any mechanistic way given the information that exists at present. The Discussion is very hard to read. Perhaps this is inevitable given the confusing nature of the results.

The calcium-dependence of PHP in the Pi3K68D mutant is surprising. As stated above, we devote a considerable portion of our Discussion section to this issue. We start the second paragraph of the Discussion as follows: “Any model to explain the role of PI3K-cII and Rab11-dependent endosomal signaling during homeostatic plasticity must account for the phenotypic observation that PHP is only blocked at low extracellular calcium concentrations.” As requested, we attempted to improve the clarity of our Discussion section, and we have shortened this section according to the request of one of the reviewers.

17) Related to point 16), they seem to be saying that since PHP works normally in the mutant at physiological (1.5 uM) Ca++ concentrations (subsection “Loss of Pi3K68D renders the expression of PHP sensitive to changes in extracellular calcium”, second paragraph), these mutants must have been normal with respect to this phenomenon during their development in vivo. It is only when they are filleted and incubated at low Ca++ that a phenotype is observed. It would be very informative about what is going on if they could determine whether the development of the larva in the absence of Pi3K68D renders fillets from these animals abnormal with regard to PHP at low Ca++, or if the activity of Pi3K68D during the experiment is required for normal PHP at low Ca++. It seems like this should be addressable in their system, because there are many Pi3K inhibitors that have been described. They could fillet wt embryos, or embryos lacking one copy of the gene, and then incubate them with the inhibitors and see if this affects PHP. But, perhaps the available inhibitors are not specific enough for this form of Pi3K, or perhaps they don't work in Drosophila? It seems like the investigators probably know the answer to this question already, since it is an obvious experiment. If so, they should inform us in the paper. If not, they should do the experiment.

This is an issue that we have made many attempts to address. Unfortunately, existing pharmacology is not applicable to *Drosophila* (and we have consulted closely with the laboratory of Dr. Kevan Shokat, a world expert on kinase signaling). Currently, there are no commercially available, specific class II PI3K inhibitors. Neither wortmannin nor LY294002 inhibit *Drosophila* class II PI3K. Thus, we moved on and attempted to inhibit Vps34 (Bago et al., 2014; Ronan et al., 2014). Vps34 inhibitors can cause a decrease in FYVE-GFP levels in mammalian cells in culture. Based on this assay, in *Drosophila* muscle, these inhibitors have no effect. These experiments included multiple concentrations of the inhibitor and tests using multiple incubation times. These reagents also do not alter PHP. Finally, it is worth noting that the active site of *Vps34*, comparing fly and mammals, is not very well conserved, providing and explanation for why the inhibitors are ineffective in *Drosophila* (Ronan et al., 2014).

18) If inhibitors don't work or are not specific enough, there is another approach that could address this question. This experiment would require more time than is normally given for revisions, and the paper is already strong, so I would not require it for publication. The Bellen lab (Nagarkar-Jaiswal et al. eLife 2015) has shown that the DeGradGFP method can transiently inhibit gene function by degrading a GFP fusion of the protein. There is a MiMIC in an appropriate position to allow replacement of the MiMIC by GFP using their methods, to create a "protein trap". In the Bellen paper, they show that such protein traps often retain function. Thus, if they were to make a protein trap and found that this allele behaved like the wt protein, they could then turn off Pi3K68D activity using the temperature-sensitive DeGradGFP mechanism in larvae that had developed with normal activity. In the Bellen paper, they show that they can rapidly shut off Dunce and turn it back on by temperature shifts.

We agree that this method could be used to acutely inhibit gene function of *Pi3K68D* and that it would be a worthwhile future experiment. We have had mixed results using this approach with other signaling systems, but it seems superior to methods of protein destruction based on oxygen radical generating reagents such as Mini-SOG and FlAsH-FALI.

19) From the images, it’s clear there are robust changes in postsynaptic endosomal compartments. However, it is less clear if there is any change going on presynaptically. Clearly they demonstrate the effect on homeostatic plasticity is postsynaptic, but it would be interesting to know if the same changes in subcellular markers is occurring in the presynaptic bouton. I assume PI(3)P would be required in both compartments to properly generate that endosomal compartment – if not it would be interesting, but very surprising. I assume the authors already have that data from their images, so would be nice to comment on it in the text.

We did quantify staining intensity using a region of interest delineated by presynaptic markers. However, upon consideration of these data, we decided against inclusion in our manuscript because it is impossible to distinguish between pre- and post-synaptic labeling as they are superimposed. Thus, it is not possible to draw any conclusions regarding pre- versus postsynaptic labeling. Ultimately, as in prior papers, it would be necessary to devise a way to exclusively isolate presynaptic endogenous expression, which is both non-trivial and not the focus our study.

20) The authors nicely show postsynaptic rescue of the Pi3K mutant phenotype. However, it wasn't clear why they didn't also show postsynaptic RNAi knockdown blocks PHP. Did I miss that somewhere – if not, why not show that as well to complement the rescue?

We identified *Pi3K68D* based on tissue specific RNAi in our genetic screen. We used additional reagents to confirm and extend this initial finding.

21) The last minor comment deals with the Pi3K68D structure function part. They infer the clathrin binding domain and kinase activity are important by overexpressing constructs missing these elements and observing a block in PHP. I would have preferred to see this being done as a rescue of the null mutant – seems a far easier result to interpret. Why was the overexpression used instead of rescue?

The rescue experiment also would have a confound in that the presynaptic neuron would lack *Pi3K68D*. The fact that there is dominant interference, with otherwise normal *Pi3K68D* expressed presynaptically, underscores the postsynaptic function of *Pi3K68D*.

22) Not sure how the EM was quantified, or if it was.

The EM figure reports an observation based on EM-tomography. It is not quantified.